# An Aurora B-RPA signaling axis secures chromosome segregation fidelity

Poonam Roshan[1,5], Sahiti Kuppa[2,5], Jenna R. Mattice[3], Vikas Kaushik[2], Rahul Chadda[2], Nilisha Pokhrel[4], Brunda R. Tumala[2], Aparna Biswas[1], Brian Bothner [3], Edwin Antony [2] ✉ & Sofia Origanti [1] ✉

Errors in chromosome segregation underlie genomic instability associated with cancers. Resolution of replication and recombination intermediates and protection of vulnerable single-stranded DNA (ssDNA) intermediates during mitotic progression requires the ssDNA binding protein Replication Protein A (RPA). However, the mechanisms that regulate RPA specifically during unperturbed mitotic progression are poorly resolved. RPA is a heterotrimer composed of RPA70, RPA32 and RPA14 subunits and is predominantly regulated through hyperphosphorylation of RPA32 in response to DNA damage. Here, we have uncovered a mitosis-specific regulation of RPA by Aurora B kinase. Aurora B phosphorylates Ser-384 in the DNA binding domain B of the large RPA70 subunit and highlights a mode of regulation distinct from RPA32. Disruption of Ser-384 phosphorylation in RPA70 leads to defects in chromosome segregation with loss of viability and a feedback modulation of Aurora B activity. Phosphorylation at Ser-384 remodels the protein interaction domains of RPA. Furthermore, phosphorylation impairs RPA binding to DSS1 that likely suppresses homologous recombination during mitosis by preventing recruitment of DSS1-BRCA2 to exposed ssDNA. We showcase a critical Aurora B-RPA signaling axis in mitosis that is essential for maintaining genomic integrity.

Maintaining genomic integrity during chromosome replication, condensation, and segregation relies on regulatory mechanisms that are distinct to each phase of the cell cycle[1,2]. Protection of transiently exposed ssDNA throughout the cell cycle is achieved through binding of Replication Protein A (RPA)[3]. RPA also serves as a protein-interaction hub to recruit other proteins onto DNA and coordinates almost all DNA metabolic processes including replication, repair, recombination, and telomere maintenance[4–6]. RPA performs several essential functions in the cell. It binds to ssDNA with high affinity ($K_D < 10^{-10}$ M) and protects it from degradation by exo- and endonucleases[3]. RPA-ssDNA complexes are important for the activation of ATR signaling response and

for a mode of double-strand break repair triggered by ATM[5–7]. RPA physically interacts with over three dozen DNA processing enzymes and recruits them to the site of DNA metabolism[4,8–10]. RPA also hands-off the DNA to these enzymes and correctly positions them on appropriate chromosomal structures to facilitate their catalytic activity[11–13]. In addition, several new cell cycle-specific functions have also been recently uncovered. For example, RPA activates a mitosis-specific R-loop driven ATR pathway for faithful segregation of chromosomes[14]. In concert with RAD52, RPA facilitates mitotic DNA synthesis (MiDAS) to counteract DNA replication stress at common fragile sites loci on the chromosomes[15,16].

[1]Department of Biology, St. Louis University, St. Louis, MO 63103, USA. [2]Department of Biochemistry and Molecular Biology, St. Louis University School of Medicine, St. Louis, MO 63104, USA. [3]Department of Biochemistry, Montana State University, Bozeman, MT 59717, USA. [4]Department of Biological Sciences, Marquette University, Milwaukee, WI 53217, USA. [5]These authors contributed equally: Poonam Roshan, Sahiti Kuppa. ✉e-mail: edwin.antony@health.slu.edu; sofia.origanti@slu.edu

To coordinate such diverse functions, RPA utilizes a unique structural assembly of DNA binding domains (DBDs) and protein interaction domains (PIDs) situated across three subunits - RPA70, RPA32, and RPA14[17–19]. There are six oligonucleotide/oligosaccharide binding (OB) folds labeled A-F (Fig. 1a). Four OB-folds (DBDs–A, B, C, and D) contribute most to ssDNA interactions. DBDs-A, B, and C are situated in the large RPA70 subunit and are connected by flexible linkers. DBD-D resides in RPA32. The heterotrimer is held together through extensive physical interactions between DBD-C, DBD-D and the RPA14 subunit (trimerization core; Fig. 1a). There are two PIDs; one is OB-F situated at the N-terminus of RPA70 (PID[70N]) and connected to DBD-A through a disordered 80 aa linker. The other PID is a winged helix (wh) domain located at the C-terminus of RPA32 (PID[32C]) and connected to DBD-D by a 34 aa disordered linker. An N-terminal ~40 aa region in RPA32 is extensively phosphorylated by a slew of kinases (Fig. 1a)[20–29]. The flexible linkers allow the domains of RPA to form various configurations (defined as the relative positions of the DBDs and PIDs) and the prevailing hypothesis is that one or more of these configurations drive specific DNA metabolic roles[4,8,10,11,30–35]. The formation of specific configurations and associated functions are thought to be regulated by post-translational modifications of RPA including phosphorylation, acetylation, sumoylation, and ubiquitination[4,36].

Post-translational modifications of RPA, especially phosphorylation of the RPA32 subunit, have been shown to regulate RPA functions based on the physiological context. RPA is phosphorylated during the S- and mitotic phases by cyclin-dependent protein kinases (CDK1/2)[14,37,38] and is hyperphosphorylated upon DNA damage by the PI3K-like family of kinases including DNA-PK[23,39–42], ATM[20,42–44], and ATR[7,14,21,42,44–47]. Phosphorylation of RPA32 has been shown to tune the timing and specificity of RPA association with factors involved in DNA replication versus DNA repair[24,42]. Most studies of RPA phosphorylation have focused on RPA32, and the characterized sites map to a 40-amino acid disordered region in the N-terminus of RPA32 (Fig. 1a). Interestingly, phosphorylation in this region modestly influences the DNA binding properties[28,29,48]. Cells carrying RPA32 lacking the phosphorylation motif are hypersensitive to DNA-damaging agents and produce mutator and hyper-mutator phenotypes[49]. Hyperphosphorylated RPA also inhibits DNA resection by blocking the Blooms helicase[50,51]. The hyperphosphorylated region of RPA32 is not part of the DBDs or PIDs (Fig. 1a). Nevertheless, the precise nature of the domain configurations and how they are modified by hyperphosphorylation remains a mystery.

While phosphorylation of RPA32 is well characterized, and often used as a marker to define replication and repair events, the effects of such modifications on other RPA subunits are not well characterized.

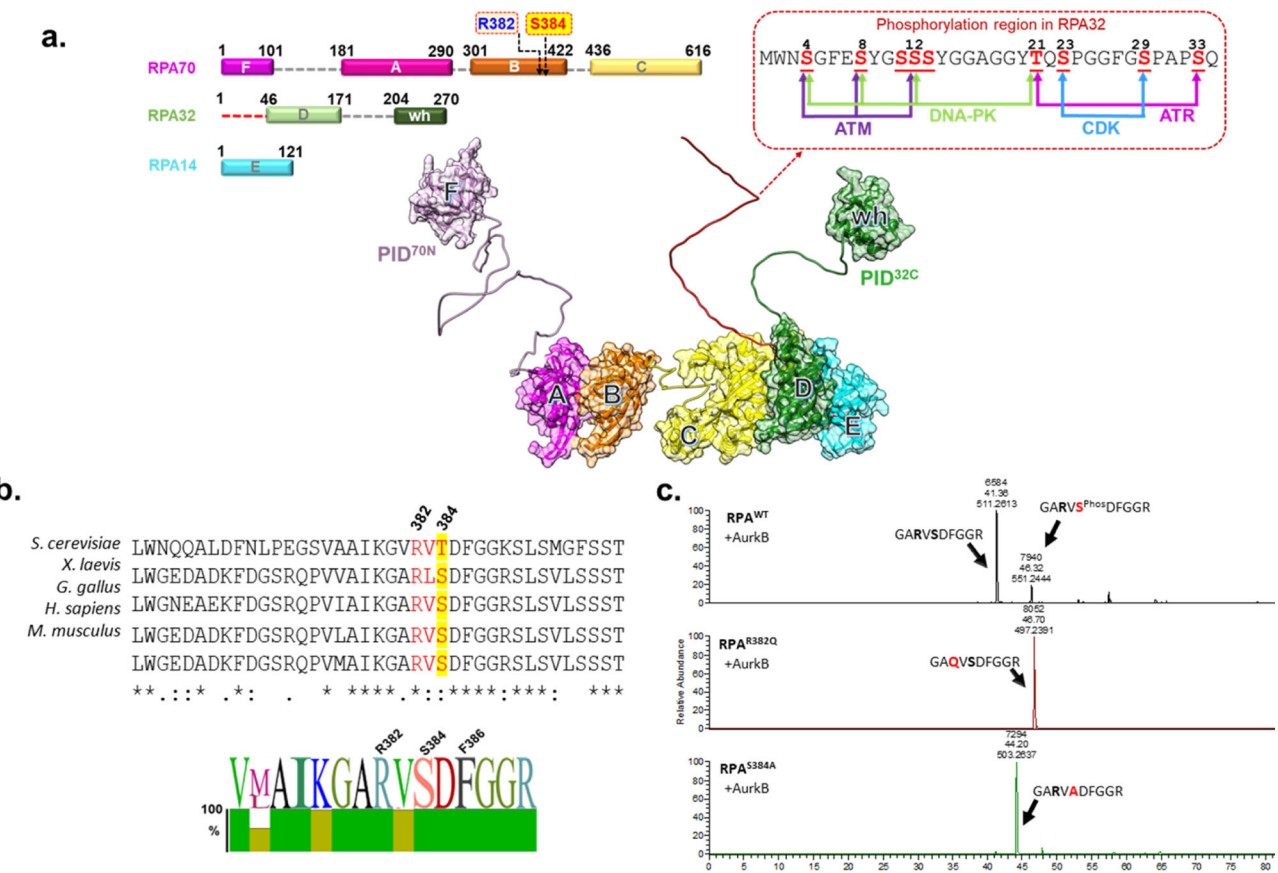

**Fig. 1 | DNA binding domain B (DBD-B) in RPA70 is phosphorylated by Aurora B kinase. a** Three RPA subunits RPA70, RPA32, and RPA14 form a heterotrimer and harbor multiple oligosaccharide/oligonucleotide (OB) domains. A, B, C, and D are DNA binding domains (DBDs). OB-F and the wh-domain are two protein interaction domains. The N-terminus of RPA32 (shown in red square) is hyperphosphorylated by multiple kinases including ATM, DNA-PK, CDK, and ATR and the known sites of phosphorylation are denoted in the insert. Arg-382 and Ser-384 in the putative Aurora B kinase motif are also noted in RPA70. The structure of human RPA OB-domains and connecting linkers were generated using AlphaFold and aligned to the trimerization core observed in the crystal structure (PDB:1L1O). **b** Sequence alignment of residues in DBD-B reveal a conserved putative phosphorylation motif for Aurora kinase in eukaryotes. A sequence logo representation of the conservation is also shown. **c** In vitro phosphorylation of recombinant RPA with Aurora B, followed by MS-MS analysis, shows residue Ser-384 in RPA70 as the sole site of phosphorylation. Extracted Ion Chromatograms (EICs) for unphosphorylated and phosphorylated tryptic peptide containing Ser-384 in the WT-RPA70 along with EIC of corresponding point mutations are shown. Alanine substitution of the Ser-384 phosphosite, or perturbation of the Aurora B recognition motif through a cancer-associated Gln substitution at Arg-382, results in loss of phosphorylation.

In *S. cerevisiae*, Ser-178 in RPA70 is phosphorylated by Mec1[43,52,53] and we recently showed that a phosphomimetic substitution in yeast RPA promotes cooperative binding of RPA molecules on ssDNA[36]. In addition, the role of RPA in DNA metabolism in the S-phase has been well characterized, but its function and regulation in mitosis, specifically during an unperturbed phase, are not well defined. Here, we have uncovered a novel mode of regulation of RPA70 that is specific to the mitotic phase of cell cycle. We show that Aurora B kinase phosphorylates Ser-384 in DNA binding domain B (DBD-B) of RPA70 in mitosis. We have also uncovered a unique feedback circuit wherein loss of RPA70 phosphorylation affects Aurora B activity. The adjacent Arg-382 residue is an integral part of the Aurora B recognition motif and also lies within the binding interface for DSS1-BRCA2[54]. Biochemically, an RPA[S384D] phosphomimetic mutant reveals configurational changes in the DBDs and releases the PIDs to promote protein-protein interactions. In addition, interactions with DSS1 are perturbed upon phosphorylation at Ser-384, thus resulting in likely loss of DSS1-BRCA2 recruitment and suppression of homologous recombination during mitosis. Loss of phosphorylation results in increased cell death, activation of p53 checkpoint response, defects in chromosome segregation, and increased sensitivity to DNA damage. Detailed functional and mechanistic characterization reveal an RPA-Aurora B signaling axis that functions through phospho-modification of RPA70 at Ser-384 during mitosis and is essential to maintain chromosome segregation fidelity.

## Results

Studies on post-translational modifications of RPA, especially phosphorylation, have focused on the N-terminal region of RPA32 (Fig. 1a). Since three of the key DNA binding domains (DBDs-A, B, and C) and a major protein interaction domain (PID[70N]) reside in RPA70, knowledge of how post-translational modifications influence these domains and RPA function is essential. Several global phosphoproteomic studies have identified Ser-384 in DBD-B of RPA70 as a prominent site of phosphorylation[55,56]. Sequence analysis of this region reveals the presence of a highly conserved minimal consensus motif $[(R/K)_{1-3}$-X-(S/T)] for Aurora B kinase[57,58] in eukaryotes (Fig. 1b). Furthermore, Arg-382 in this motif has been identified as part of the RPA interaction site for BRCA2-DSS1[54] and thus warrants detailed functional characterization.

### Aurora B kinase phosphorylates RPA in vitro

To test whether RPA is phosphorylated by Aurora B, we performed in vitro kinase assays and analyzed modifications in the RPA subunits using mass spectrometry and autoradiography. Site-specific phosphorylation of RPA70 is observed only in the presence of Aurora B and a single site of phosphorylation at Ser-384 is identified in MS/MS analysis (Fig. 1c). Mutation of Ser-384 to Ala results in loss of phosphorylation (Fig. 1c and Supplementary Fig. 1a, b). Arg-382 in the Aurora B consensus motif is also important as Arg-382 to Gln mutations have been observed in primary skin and thyroid cancers, albeit at low frequency (COSMIC)[59]. An Arg-382 to Gln substitution in the motif (RPA[R382Q]) also abolishes phosphorylation by Aurora B as shown by MS analysis and autoradiography (Fig. 1c and Supplementary Fig. 1). Faint background signal is observed for RPA32 and both mutants of RPA70. However, MS analysis did not identify other sites of phosphorylation in the other two subunits (RPA32 or RPA14) and for RPA70 in the two mutants, under all conditions tested. These data suggest that Aurora B kinase primarily phosphorylates the RPA70 subunit of RPA at the Ser-384 site.

### Phosphorylation of RPA70 at Ser-384 is specific to mitosis

Next, we investigated the cellular regulation of RPA by Aurora B. Since Aurora B functions in mitosis, we probed for RPA70 phosphorylation at Ser-384 (pS384-RPA70) in asynchronous and mitotic HCT116 and 293T cells using a custom-generated phospho-specific antibody. The antibody was found to be specific for the Ser-384 phosphorylated form of RPA by in vitro kinase assay and by phosphatase treatment of lysates (Supplementary Fig. S1 c, d). In both these cell lines, phosphorylation at the Ser-384 site is observed only during mitosis (Fig. 2a, b), which is lost upon entry into G1 phase. A slower migrating form of total RPA70 was also enriched in mitotic cells suggesting phospho-modification (Fig. 2a). Induction of Ser-10 phosphorylation of Histone H3 was used as a marker for mitosis. These data show that Aurora B-mediated phosphorylation of RPA is specific to mitosis, in agreement with the well-established mitotic functions of Aurora B[60].

Further assessment of pS384-RPA70 across all phases of the cell cycle by synchronization using double thymidine block shows that Aurora B phosphorylation of RPA is observed only in mitosis (Fig. 2c). HCT116 cells were efficiently synchronized at the G1/S boundary using double thymidine block as shown by flow cytometry (Supplementary Fig. 2 a–d, e). To further determine the specificity of mitotic-specific phosphorylation of RPA70 by Aurora B kinase, cells were treated with an Aurora B-specific inhibitor (AZD1152)[61,62]. Inhibition of Aurora B was confirmed by loss of Ser-10 Histone H3 phosphorylation, which is a well-characterized substrate specific for Aurora B in mitosis (Fig. 2d). Short-term treatment with the Aurora B inhibitor caused a marked inhibition of Ser-384 phosphorylation of RPA70. Knockdown of Aurora B also led to a corresponding decrease in phosphorylation of Ser-384 in RPA70 that could be rescued by overexpression of Aurora B without any changes to total RPA70 levels (Supplementary Fig. 2f, g). These results suggested that Aurora B is the kinase that predominantly phosphorylates RPA70 in mitosis at Ser-384. In addition, we found that phosphorylation at Ser-384 is not observed upon induction of replication stress by metabolite of irinotecan (SN-38; Fig. 2e). Effect of SN-38 was confirmed by stabilization of p53. These data further showed that phosphorylation at Ser-384 of RPA70 is not regulated by replication stress.

### Phosphorylation at Ser-384 of RPA70 moderately alters the ssDNA binding properties of RPA

The Aurora B motif in DBD-B of RPA70 is situated within the DNA binding pocket (Fig. 3a). In the structure (PDB: 1JMC)[63], Arg-382 makes several contacts with the backbone of ssDNA. While Ser-384 does not directly interact with the DNA in the structure, it makes backbone interactions with Phe-386, which in turn base-stacks with ssDNA and is thus critical for DBD-B interaction with DNA (Fig. 3a). To test if phosphorylation at Ser-384 in RPA70 influences the ssDNA binding properties of RPA, we measured the DNA binding activity of RPA using fluorescein-labeled $(dT)_{20}$ or $(dT)_{40}$ oligonucleotide substrates (Fig. 3b, c). In fluorescence anisotropy experiments we see no observable differences in DNA binding activities between RPA and the phosphomimetic mutant RPA[S384D] (Fig. 3b, c). Since RPA has several high-affinity DBDs, subtle differences in ssDNA binding by individual DBDs are often hidden by the overall macroscopic DNA binding effect; i.e., ssDNA binding will be observed irrespective of whether one or more DBDs are bound to DNA[11]. To better tease out subtle differences in DNA binding properties, we utilized rapid kinetic experiments to capture the rate of RPA binding to ssDNA. Using stopped flow analysis, we monitored the change in intrinsic tryptophan (Trp) fluorescence of RPA upon ssDNA binding (Fig. 3d, e). We observed changes in the rate and amplitude of Trp quenching. Plotting the observed rates of Trp fluorescence change versus DNA concentration yields $k_{on}$ and $k_{off}$ for DNA interactions (Fig. 3f). RPA[S384D] has a slower $k_{on}$ ($1.9 \pm 0.1 \times 10^{10}$ M$^{-1}$ s$^{-1}$) and faster $k_{off}$ ($65 \pm 7$ s$^{-1}$) compared to RPA ($k_{on} = 5.2 \pm 0.3 \times 10^{10}$ M$^{-1}$ s$^{-1}$ and $k_{off} = 39 \pm 4$ s$^{-1}$). Thus RPA[S384D] binds to ssDNA with ~5-fold lower affinity ($K_D = 3.4 \pm 0.2$ nM) compared to RPA ($K_D = 0.75 \pm 0.4$ nM). Introduction of the negative charge at position 384 likely influences DNA binding to DBD-B and/or the path of ssDNA binding within this domain.

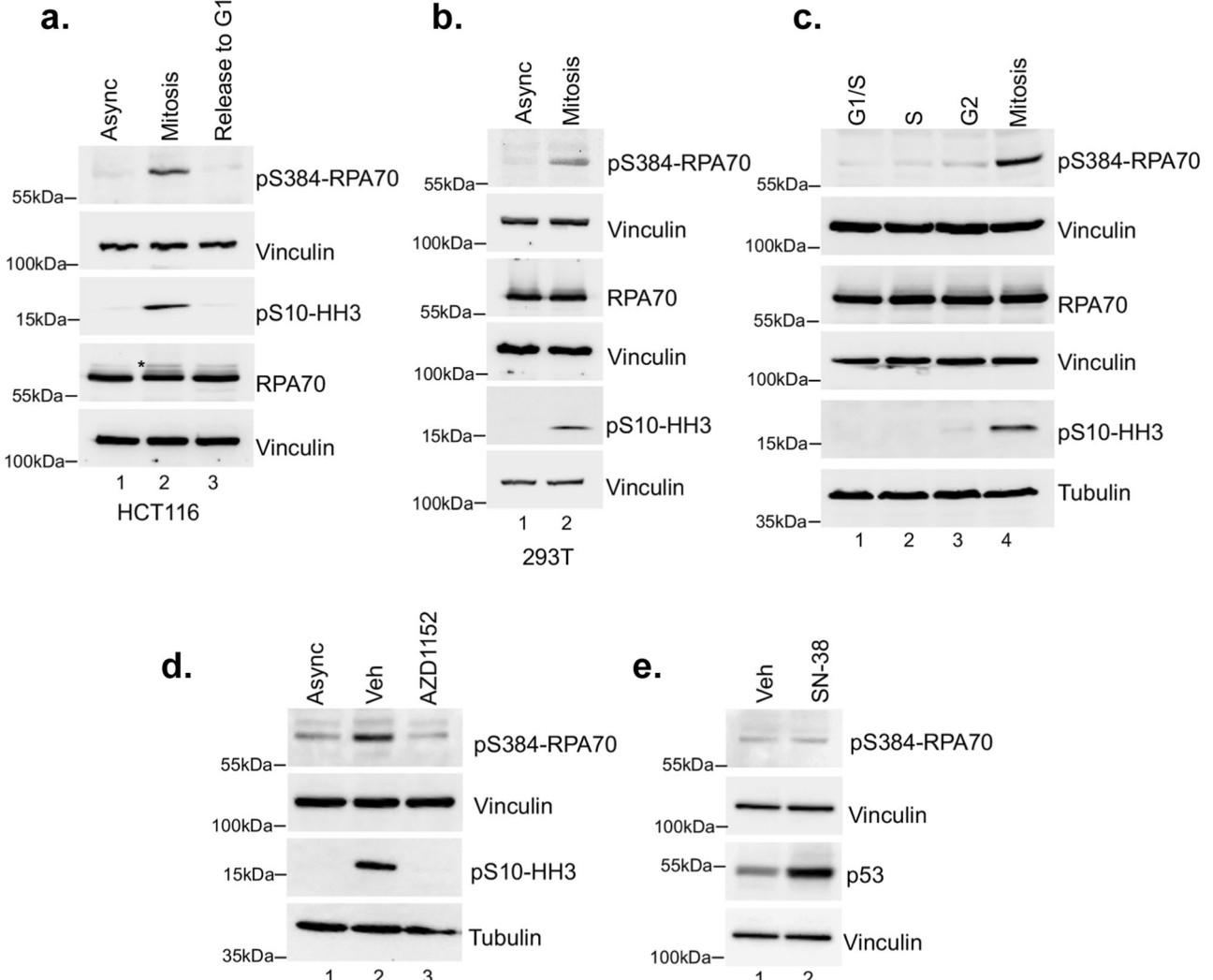

**Fig. 2 | Aurora B phosphorylates RPA at Ser-384 specifically in mitosis.**
**a** Western blot analysis of asynchronous (Async) HCT116 cells, nocodazole-arrested mitotic cells, and arrested cells released (3 h) into G1 phase. Blots were probed with monoclonal phospho Ser-384 RPA70 (pS384-RPA70) (custom antibody, Genscript) and total RPA70 antibodies. Vinculin was used as a loading control. Mitotic arrest was confirmed using phospho-Ser10-histone H3 specific antibody (pS10-HH3). * Indicates a slower migrating form of RPA70 enriched in mitotic cells. **b** Similar mitosis-specific phosphorylation of RPA70 at Ser-384 was also observed in 293 T cells using western blot analysis. All subsequent studies were carried out in HCT116 cells. **c** RPA70 phosphorylation at Ser-384 was assessed in different phases of the cell cycle by synchronization at G1/S boundary using double thymidine block followed by release into S and G2 phases. Phosphorylation in mitosis was assessed as described in a). Tubulin and vinculin were used as loading controls. Specific phosphorylation of RPA70 at Ser-384 is observed only during mitosis. **d** Aurora B was selectively inhibited in mitotic cells with short-term treatment (45 min) of 3 μM AZD1152 and results in loss of RPA70 Ser-384 phosphorylation. Inhibition of Aurora B was confirmed by loss of Ser10-Histone H3 phosphorylation. Asynchronous cells and vehicle-treated (0.1% DMSO) mitotic cells were used as controls. **e** Cells were treated with vehicle (Veh; 0.1% DMSO) or 10 ng/ml SN-38 for 21 h to induce replication stress. DNA damage does not induce RPA70 phosphorylation at Ser-384. All blots in this figure are representative of at least three independent replicates.

## Ser-384 phosphorylation of RPA70 promotes the formation of higher-density RPA binding to ssDNA

In the Trp quenching experiments we noticed a significant difference in the basal intrinsic Trp fluorescence between RPA and RPA$^{S384D}$ (Supplementary Fig. 3a). Upon ssDNA binding, the amplitude of Trp fluorescence quenching was similar, but the signals did not reach the same plateau. Interestingly, the measurement of secondary structure using circular dichroism (CD) shows no difference between RPA and RPA$^{S384D}$ (Supplementary Fig. 3b). Thus, the phosphomimetic substitution does not change the folding of DBD-B or other neighboring domains. We hypothesized that the change in intrinsic Trp fluorescence likely originates from altered configuration(s) of the DBDs (orientation/position of DBD-B with respect to the other domains) upon phosphorylation by Aurora B. Changes in RPA configuration promote formation of nucleoprotein filaments that are structurally

different. For example, in *S. cerevisiae* RPA, a phosphomimetic substitution at Ser-178 situated just outside of DBD-A promotes cooperative binding of RPA to ssDNA[36]. Here, the DBD-A from one RPA molecule interacts with OB-E of the neighboring RPA and these interactions are proposed to be stabilized upon phosphorylation. Since we observe evidence for configurational changes in RPA upon Aurora B phosphorylation at DBD-B, we tested whether binding of multiple RPA molecules on short versus long ssDNA substrates is altered. Binding of RPA or RPA$^{S384D}$ to short (dT)$_{35}$ or longer (dT)$_{97}$ ssDNA oligonucleotides were analyzed using size exclusion chromatography (SEC; Supplementary Fig. 4). On the shorter DNA substrate, a single peak corresponding to one RPA bound to ssDNA is observed for RPA (Supplementary Fig. 4a). In contrast, for RPA$^{S384D}$, an additional larger species is observed suggesting binding of multiple RPA molecules (Supplementary Fig. 4a). This phenomenon is exaggerated on the

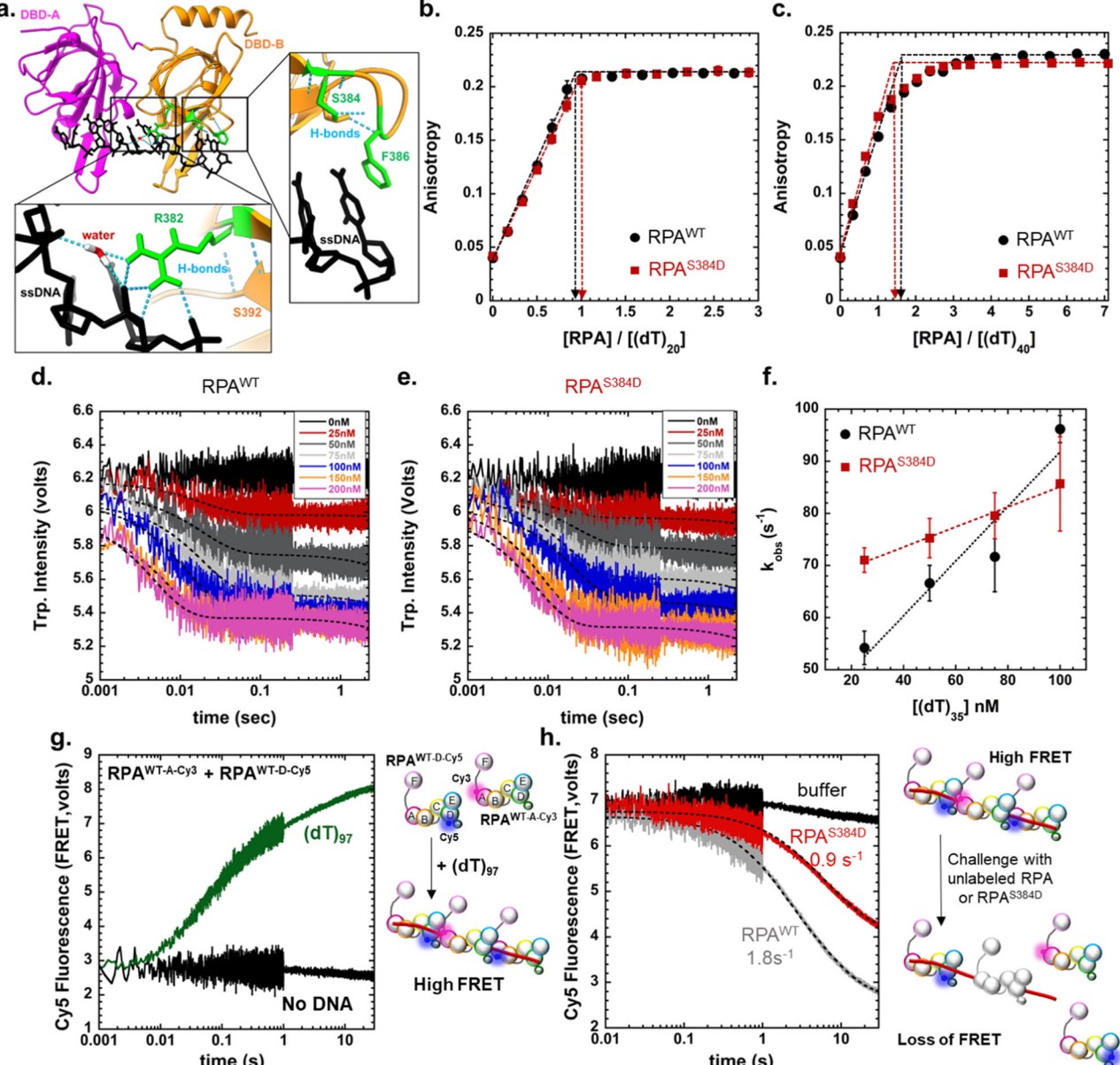

**Fig. 3 | A phosphomimetic S384D substitution minimally influences the DNA binding properties of RPA but induces configurational changes. a** Contacts between Arg-384 and the ssDNA are shown in the crystal structure of DBD-A, B (PDB:1JMC). Ser-384 does not directly contact the ssDNA, but positions Phe-386 to promote a key base stacking interaction with the base. RPA or RPA$^{S384D}$ binding to DNA was measured by fluorescence anisotropy using either (**b**) 5'-FAM-(dT)$_{20}$ or (**c**) 5'-FAM-(dT)$_{40}$ ssDNA oligonucleotides. In both cases, high-affinity stoichiometric binding is observed for RPA and RPA$^{S384D}$ and no significant changes in the binding behavior are observed. Stopped flow kinetic analysis of ssDNA binding was performed with (**d**) RPA or (**e**) RPA$^{S384D}$ by monitoring the change in intrinsic tryptophan fluorescence as a function of ssDNA (dT)$_{35}$ concentration. **f** Plot of the $k_{obs}$ versus ssDNA concentration yields $k_{on}$ and $k_{off}$ values. RPA$^{S384D}$ has a slower $k_{on}$ ($1.9 \pm 0.1 \times 10^{10}$ M$^{-1}$ s$^{-1}$ and faster $k_{off}$ ($65 \pm 7$ s$^{-1}$) compared to RPA$^{WT}$ ($k_{on} = 5.2 \pm 0.3 \times 10^{10}$ M$^{-1}$ s$^{-1}$ and $k_{off} = 39 \pm 4$ s$^{-1}$). $K_D$ values extracted from these

measurements show RPA$^{WT}$ binding to ssDNA with ~5-fold higher affinity ($K_D = 0.75 \pm 0.4$ nM) compared to RPA$^{S384D}$ ($K_D = 3.4 \pm 0.2$ nM). **g** A Förster Resonance Energy Transfer (FRET) experiment was developed using two fluorescent versions of RPA. RPA was site-specifically labeled on either DBD-A with Cy3 or DBD-D with Cy5. Equimolar ratios of both fluorescent RPA were mixed with ssDNA (dT)$_{97}$ in a stopped flow. Changes in Cy5 fluorescence were monitored by exciting Cy3 at 535 nm. Assembly of multiple RPA on the long ssDNA substrate results in a high FRET signal (green trace). In the absence of ssDNA, no enhancement in fluorescence is observed. **h** RPA filaments were pre-formed on ssDNA (dT)$_{97}$ using the Cy5- and Cy3-labeled RPA and facilitated exchange activity was measured by challenging the RPA$^{Cy3}$-RPA$^{Cy5}$-ssDNA assembly with mixing against unlabeled RPA or RPA$^{S384D}$. RPA exchanges the fluorescent-RPA faster ($k_{FE} = 1.8$ s$^{-1}$) compared to RPA$^{S384D}$ ($k_{FE} = 0.9$ s$^{-1}$). Data are presented as ±SEM from three independent experiments.

longer ssDNA substrate where a higher number of RPA$^{S384D}$ molecules are bound compared to RPA (Supplementary Fig. 4b). Thus, we propose that phosphorylation of RPA at Ser-384 is changing the arrangement of DBD-B and this configurational change promotes higher density of RPA molecules bound on ssDNA.

Such configurational changes could affect the stability of the RPA nucleoprotein filament and alter the accessibility of ssDNA. We followed the facilitated exchange (FE) activity of RPA as an experimental measure of filament stability. During FE, RPA bound on ssDNA is replaced by free RPA in solution[32,64]. While the mechanisms underlying FE are poorly understood, it is thought that the dynamic binding and dissociation of the individual DBDs allow RPA to exchange. Since phosphorylation by Aurora B mildly affects the DNA binding properties of DBD-B, we tested FE on nucleoprotein filaments formed by

either RPA or RPA$^{S384D}$. Mixing equimolar concentrations of RPA-DBD-A$^{Cy3}$ and RPA$^{WT}$-DBD-D$^{Cy5}$ with (dT)$_{97}$ ssDNA results in a FRET-induced increase in Cy5 fluorescence when Cy3 is excited (Fig. 3g, h). The FRET signal arises from the defined polarity of RPA-ssDNA interactions where DBD-A resides towards the 5′ end of the DNA[65]. Thus, DBD-A from one RPA molecule sits adjacent to DBD-D of the neighboring RPA. To measure FE, we premixed RPA$^{WT}$-DBD-A$^{Cy3}$, RPA$^{WT}$-DBD-D$^{Cy5}$ and (dT)$_{97}$ and a corresponding high FRET signal is observed (Fig. 3g). When this filament is challenged with either unlabeled RPA or RPA$^{S384D}$, loss of the FRET signal is observed as the fluorescent RPA molecules are replaced by the unlabeled RPA or RPA$^{S384D}$ during FE. RPA is more efficient in performing FE ($k_{FE}$ = 1.8 s$^{-1}$) compared to RPA$^{S384D}$ ($k_{FE}$ = 0.9 s$^{-1}$; Fig. 3h). Thus, phosphorylation at Ser-384 and the resulting change in configuration of DBD-B enables easier remodeling of RPA nucleoprotein filaments.

### Loss of Ser-384 phosphorylation of RPA70 markedly affects cell viability

To determine the physiological significance of configurational changes in DBD-B induced by Ser-384 phosphorylation, we generated a homozygous knock-in of Ser-384-Ala RPA70 mutant in the endogenous loci in HCT116 cells. As expected, the phospho-dead mutant (RPA$^{SA/SA}$) does not show phosphorylation at the Ser-384 site in mitosis (Fig. 4a). This finding further validates the specificity of the antibody used. In addition, the phospho-dead mutation does not affect the endogenous levels of total RPA70 (Fig. 4a). To investigate the functional effects of Ser-384 phosphorylation, we first investigated cell viability using MTS assays. Cell viability is markedly reduced in the RPA$^{SA/SA}$ mutant relative to the isogenic parental wild type (RPA$^{WT/WT}$) cells (Fig. 4b), which is consistent with morphological assessments that indicate increased cell death in the RPA$^{SA/SA}$ mutant. We also observed similar loss of viability in a second clone of RPA$^{SA/SA}$ mutant (Supplementary Fig. 5). Significantly higher rates of apoptosis in the RPA$^{SA/SA}$ mutant were further confirmed by measuring Caspase-3 and Caspase-7 enzymatic activity using the Caspase-3/7 glo assay (Fig. 4c). The phospho-dead mutant was also subjected to replication stress induced by SN-38 (Fig. 4d) and showed enhanced sensitivity to replication stress (Fig. 4e). Analysis of asynchronous cells did not reveal any differences in the percentage of S-phase cells that could account for increased sensitivity of the phospho-dead mutant to replication stress. A 3–4% decrease in G1 population of the phospho-dead mutant was observed (Supplementary Fig. 6a–c). Interestingly, the phospho-dead mutant displayed high basal levels of p53 even in the absence of SN-38 treatment indicating that the mutant cells are in a constant state of genomic stress that leads to enhanced p53 stability (Supplementary Fig. 6d, e, f). Mild increase in γH2AX levels was also observed in the mutant cells (Supplementary Fig. 6d). However, unlike the p53 checkpoint response, loss of phosphorylation does not activate the Chk1-mediated checkpoint response at a basal state (Fig. 4d). In addition, the p53 and Chk1-mediated checkpoint responses were induced by replication stress to a similar extent between the WT and mutant cells (Fig. 4d). This indicates that the DNA damage-induced checkpoint response is not altered by loss of Ser-384 phosphorylation. However, loss of phosphorylation enhances basal genomic stress response and induces extensive apoptosis that further sensitizes cells to replication stress.

### Mitotic phosphorylation of RPA70 is critical for chromosome segregation

Since Ser-384 phosphorylation of RPA70 occurs specifically in mitosis, we wanted to test if defects in mitotic progression contribute to genomic instability. Mitosis is defined by the precise segregation of sister chromatids to opposite spindle poles. Defects in chromosome segregation and nondisjunction can lead to a marked increase in anaphase DNA bridges between the segregating sister chromatids[66,67]. To uncover the functional significance of mitosis-specific phosphorylation of RPA70, we assessed the mitotic phases of the phospho-dead mutant. Interestingly, both in unperturbed mitosis in asynchronous populations and in cells released from pro-metaphase arrest, there was a marked increase in chromosome segregation defects induced by loss of phosphorylation (Fig. 5 and Supplementary Fig. 7a). Segregation defects were highlighted by the enhanced presence of anaphase bridges and lagging chromosomes. We did not observe enhanced defects in spindle pole formation or spindle alignments in the phospho-dead mutant relative to WT, which indicated that the chromosome segregation defects are likely driven by unresolved replication or recombination intermediates or misalignment of chromatid kinetochore and microtubule attachments. Intriguingly, when we probed for Ser-10 phosphorylation of Histone H3 (the mitotic marker), we observed a 50% decrease in Ser-10 phosphorylation in the phospho-dead mutant (Fig. 6a, b). This change in phosphorylation of Histone H3 could not be attributed to changes in total Histone H3 levels or due to differences in mitotic synchronization as shown by flow cytometric analysis (Fig. 6c-e). Since phosphorylation of Ser10 in Histone H3 is important for chromosome condensation, although the mechanism remains poorly resolved, we compared chromosome compaction in WT and phospho-dead mutant as determined by the surface area of the nuclei (Supplementary Fig. 7b). Consistent with the decrease in Ser10-phosphorylation of Histone H3, there was less chromosome compaction of many nuclei in the phospho-dead mutant (Fig. 6f). Aurora B phosphorylates Ser10 of Histone H3 and therefore, we wanted to determine if the decrease in Histone H3 phosphorylation was due to a decrease in Aurora B activity. To determine changes in Aurora B activity, we probed for the autophosphorylation status of Aurora B at the Thr232 site (Fig. 6g, h). Intriguingly, autophosphorylation of Aurora B was reduced by about 40% in the phospho-dead mutants (Fig. 6g, h) indicating that Aurora B activity is regulated through a feedback mechanism mediated by phosphorylation of RPA (Fig. 6i). Thus, these chromosomal defects clearly indicate that Ser-384 phosphorylation of RPA70 is important for segregation of chromosomes in mitosis.

### Phosphorylation at Ser-384 changes the configuration of RPA domains

To better understand how phosphorylation by Aurora B influences the configurational changes within the multi-domain structure of RPA, we performed hydrogen-deuterium exchange mass spectrometry (HDX-MS) analysis of RPA$^{WT}$ and RPA$^{S384D}$ in the absence or presence of ssDNA (Fig. 7 and Supplementary Table 1). Configurational differences cause changes in uptake of deuterium[30]. The incorporation of deuterium over time is plotted as a comparison between RPA and RPA$^{S384D}$ (Fig. 7, and Supplementary Fig. 8–15). The impact of ssDNA binding on the uptake of the two protein forms is also shown.

Surprisingly, changes in HDX are observed in several regions including DBDs-A, B, the trimerization core, the two protein interaction domains, and the flexible linker between DBD-B and DBD-C (Fig. 7a). In the presence of ssDNA, the patterns of HDX change (Fig. 7b). However, the changes again cover multiple domains including DBDs-A, B, the trimerization core, and one of the protein interaction domains (OB-F; Fig. 7b). These changes persist over longer time scales suggesting that the configurational changes driven by the phosphomimetic substitution are stable (Supplementary Fig. 8). These data reveal that the various domains of human RPA do not exist in a splayed-out fashion, but likely interact with each other in a configurationally compacted form. These interactions are altered upon phosphorylation at Ser-384. Since the DNA binding properties are not severely affected, another functional outcome could be the modulation of protein-protein interactions between RPA and RIPs. Since OB-F in RPA70 (PID$^{70N}$) and the winged-helix domain in RPA32 (PID$^{32C}$) show

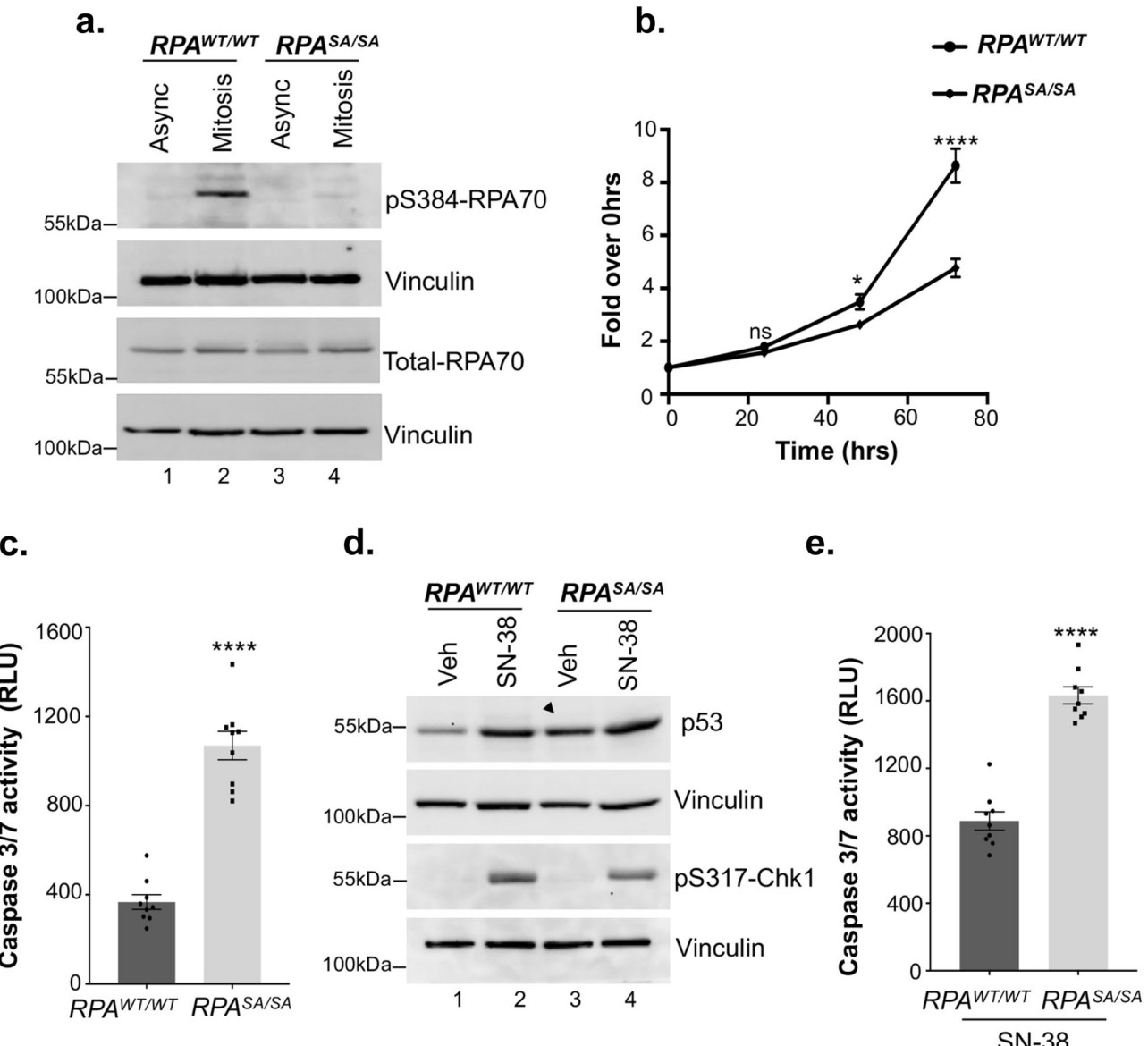

**Fig. 4 | Loss of Ser-384 RPA70 phosphorylation markedly disrupts cell viability and induces basal genomic stress response. a** Homozygous knock-in of Ser-384-Ala RPA70 phospho-dead mutant in HCT116 cells was generated using CRISPR-Cas9 editing. Cells were synchronized in mitosis using nocodazole and representative western blot depicts the loss of RPA70-Ser-384 phosphorylation in RPA $^{SA/SA}$ mutant. Asynchronous parental HCT116 cells and cells arrested in mitosis were used as controls. Blots were probed with the indicated antibodies. Blots are representative of at least three independent experiments. **b** MTS assay shows decreased viability of phospho-dead $RPA^{SA/SA}$ mutant. Cells were assayed at 0, 24, 48, and 72 h of growth. Values corrected for background absorbance were normalized to 0 h of growth. Error=SEM. The mean of three independent experiments was plotted. Triplicate wells were assayed per time point for each experiment. Statistical significance was determined using an unpaired two-tailed $t$-test: $p = 0.38$ at 24 h, ns = not significant, $*p = 0.014$ at 48 h, $****p = 0.000018$ at 72 h. **c** Caspase 3/

7 activity was significantly higher in the RPA mutant cells as determined by the Caspase3/7 glo assay. Bar graph depicts data corrected for background absorbance. Error = SEM. The mean of three independent experiments was plotted. Triplicate wells were assayed per experiment. Statistical significance was determined using an unpaired two-tailed $t$-test: $****p = 0.00000003$. **d** Representative western blot shows replication stress-response in cells treated with vehicle (veh, 0.1% DMSO) or 20 ng/mL SN-38 for 90 min. Arrow indicates increased basal genomic stress as shown by high levels of p53 in the $RPA^{SA/SA}$ mutant. Data are representative of three independent experiments. **e** Caspase3/7 activity was determined in response to replication stress by treating cells with 10 ng/mL SN-38 for 21 h and by using Caspase3/7 glo assay. Bar graph displays data corrected for background absorbance. Error = SEM. The mean of three independent experiments was plotted. Triplicate wells were assayed per experiment. Statistical significance was determined using an unpaired two-tailed $t$-test: $****p = 0.00000002$.

changes in HDX (Fig. 7a, b), phosphorylation at Ser-384 also likely releases them and promotes RPA-protein interactions through these domains.

### Ser-384 phosphorylation inhibits DSS1 binding to RPA

While PID$^{70N}$ and PID$^{32C}$ primarily coordinate most of the characterized RPA interactions with RPA-interacting proteins, a few interactors bind to other regions in RPA70. For example, physical interactions between

DSS1 and RPA have been mapped to DBD-B[54]. DSS1 works in complex with BRCA2 to facilitate the loading of RAD51 on RPA-coated ssDNA[68]. Physical interaction between DSS1 and RPA is required for this activity and the binding interface resides within the F-A-B half (domains OB-F, DBD-A, and DBD-B) of RPA. Interestingly, in NMR studies, Arg-382 is one of the residues that show a chemical shift perturbation upon DSS1 binding[54]. Arg-382 is part of the Aurora B kinase motif (Fig. 1b) and is found mutated in certain cancers. Thus, we tested whether

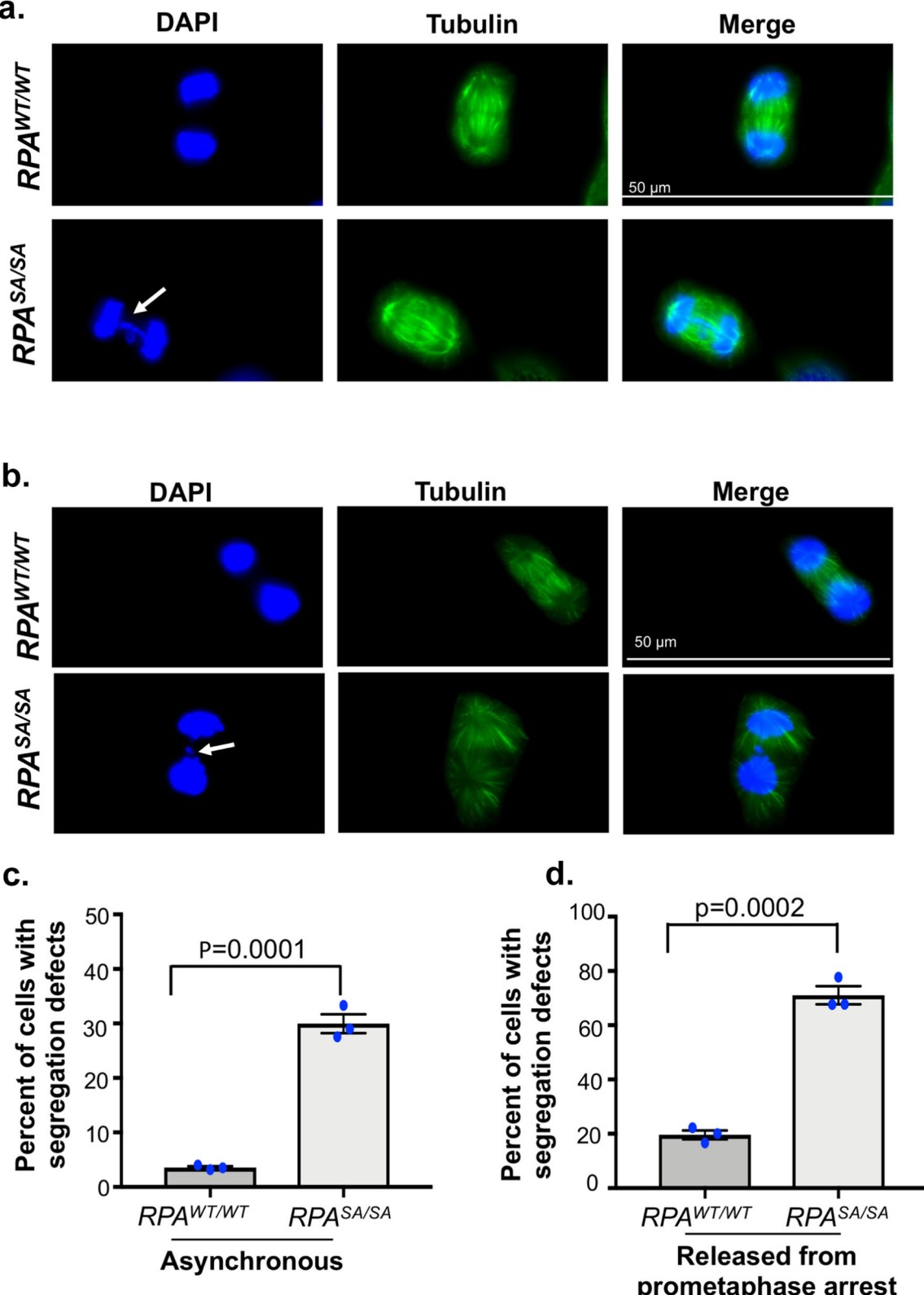

**Fig. 5 | Defects in chromosome segregation induced by loss of Ser-384 RPA70 phosphorylation.** Representative immunofluorescent images stained with DAPI and anti-Tubulin antibody depict anaphase bridges (white arrow) in the mitotic population of asynchronous *RPA^SA/SA* mutant and *RPA^WT/WT* cells (**a**) and in cells released from prometaphase arrest (**b**). Images are representative of three independent experiments. Quantitation of anaphase bridges and lagging chromosomes in mitotic population of asynchronous cells (**c**) and in cells released from prometaphase arrest (**d**) *RPA^WT/WT* and *RPA^SA/SA* cells. More than 70 cells were counted from three independent experiments. Data represent SEM and statistical significance was determined using an unpaired two-tailed *t*-test. ***$p = 0.0001$ (**c**) and *** $p = 0.0002$ (**d**).

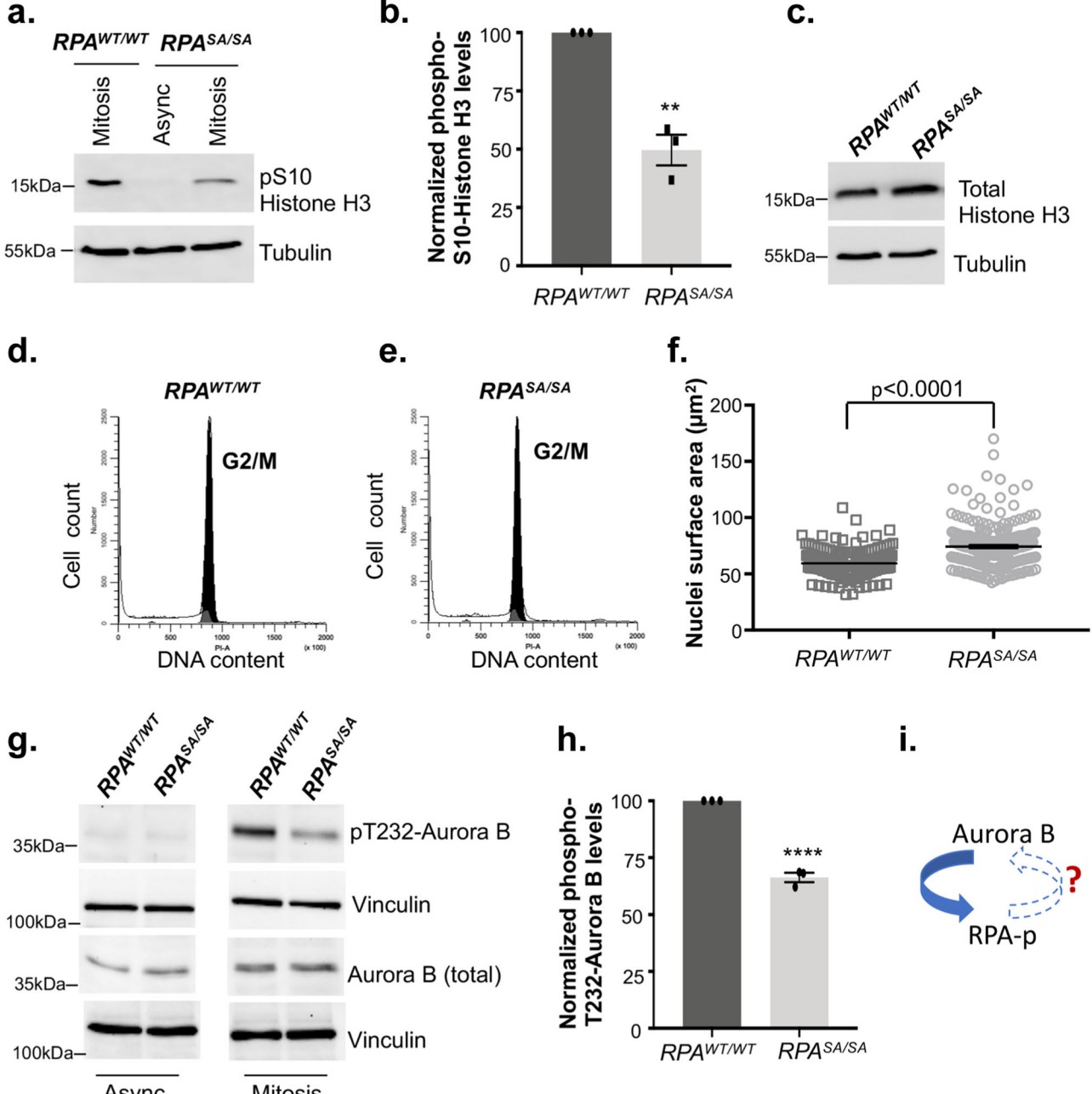

**Fig. 6 | The phospho-dead *RPA^SA/SA* mutant exhibits reduced phosphorylation of Histone H3 and Aurora B. a** Western blot represents a decrease in Ser10 Histone H3 phosphorylation in *RPA^SA/SA* mutant relative to wild type cells. Asynchronous (async) cells were assayed as control. Blots are representative of three independent experiments. **b** Blots represented in (**a**) were quantitated and normalized to loading control (Tubulin). Data are expressed as a fold-over *RPA^WT/WT*. Data are presented as mean of three independent experiments and error = SEM. Statistical significance was determined using an unpaired two-tailed *t*-test: **\*\*p* = 0.0016. **c** Western blot represents total Histone H3 levels in *RPA^SA/SA* mutant and parental cells. Blots are representative of three independent experiments. *RPA^WT/WT* (**d**) cells synchronized in mitosis and collected by shake-off method were found to be arrested in G2/M phase of cell cycle similar to *RPA^SA/SA* mutant (**e**) as analyzed by flow cytometry. DNA content was determined using propidium iodide staining. Data represent three independent experiments. **f** Surface area of nuclei stained for phospho Ser-10

Histone H3 represented in supplementary Fig. 7b. were quantitated using Image J analysis. Data are presented as a scatter plot of more than 200 nuclei measured across three independent measurements. Error = SEM. Statistical significance was determined using an unpaired two-tailed *t*-test. *p* = 0.00000000001. **g** Blots represent the changes in Aurora B kinase activity as determined by T232 autophosphorylation and total levels. Vinculin = loading control. **h** Plot depicts decreased Aurora B activity in the *RPA^SA/SA* mutant relative to WT in mitotic cells. Asynchronous cells were used as controls. Aurora B-T232 phosphorylation levels were normalized to the loading control. Data are representative of three experiments. Error=SEM. Statistical significance was determined using an unpaired two-tailed *t*-test: **\*\*\*\*p* = 0.000087. **i** Schematic shows the feedback regulation of Aurora B activity through Ser-384 RPA70 phosphorylation in mitosis. However, the precise mechanism of regulation remains to be determined.

phosphorylation in this motif interferes with DSS1 binding using bio-layer interferometry. Strep-tagged DSS1 was tethered onto a streptavidin-coupled optical probe and binding to RPA^WT or RPA^S384D was measured. A reduction in binding to DSS1 is observed for RPA^S384D

(Fig. 7c). Since DSS1-BRCA2 would encounter RPA-ssDNA structures in the cell, we tested the binding interaction when RPA is in complex with ssDNA. In this context, complete loss of binding to DSS1 is observed for RPA^S384D (Fig. 7d). Thus, Ser-384 phosphorylation by Aurora B perturbs

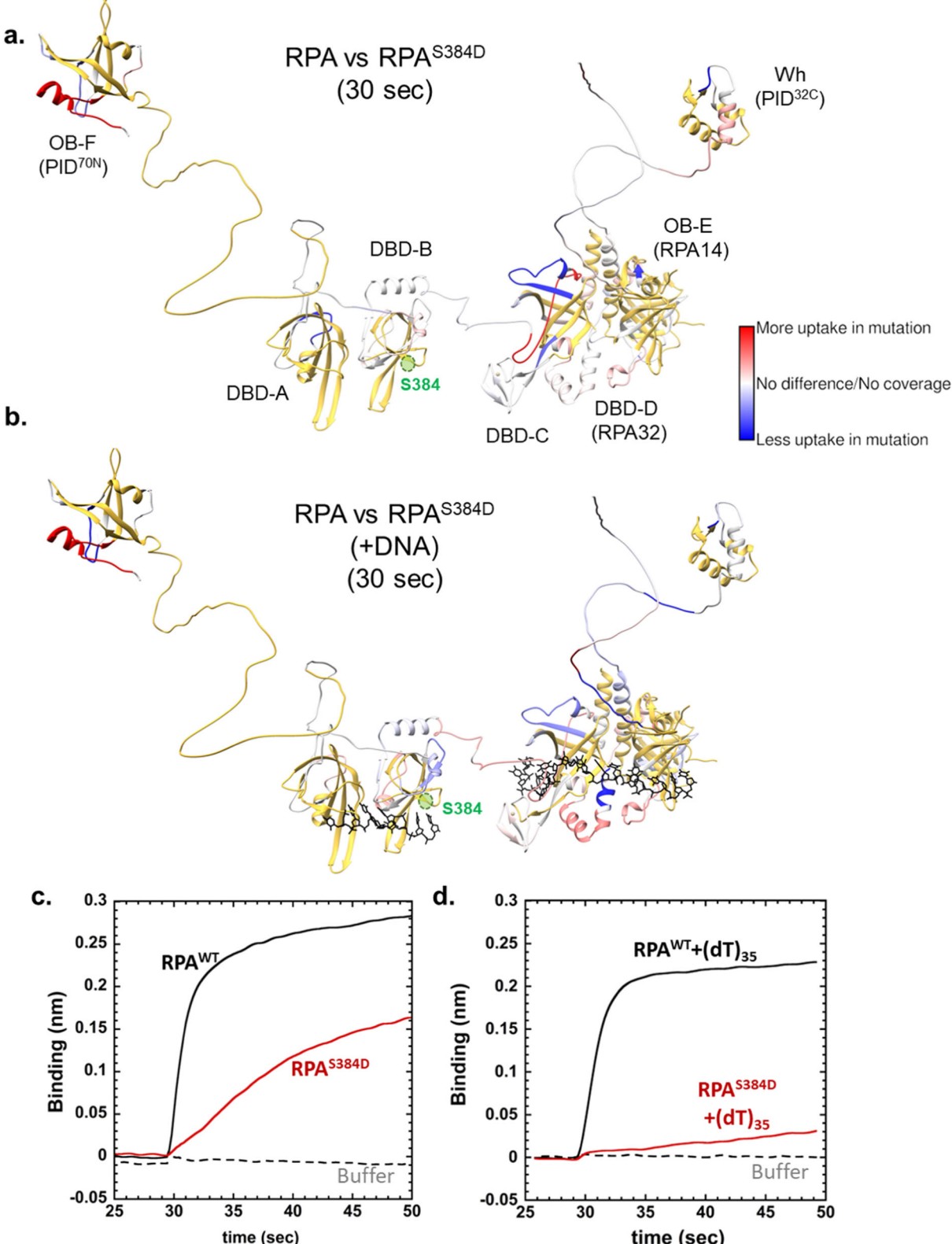

**Fig. 7 | Configurational changes in RPA are induced by a S384D substitution within the Aurora B motif in DBD-B.** HDX changes between RPA and RPA$^{S384D}$ are shown in the (**a**) absence or (**b**) presence of ssDNA (ssDNA is depicted in black). Changes in deuterium incorporation are observed in almost all DNA binding and protein-interaction domains. Data are mapped onto the structure of human RPA which is built using the structures of the OB domains from crystal structures. The regions colored yellow correspond to peptides that were not identified in the MS analysis of either or both the wild type and mutant RPA samples. The flexible linkers were modeled using AlphaFold. The position of Ser-384 is denoted in green. Data are presented as ±SDM from three independent experiments. Bio-layer interferometry analysis of RPA or RPA$^{S384D}$ binding to DSS1 in the (**c**) absence or (**d**) presence of ssDNA. RPA$^{S384D}$ shows reduced binding to DSS1 in the absence of DNA. When ssDNA-bound, almost complete loss of DSS1 binding to RPA is observed.

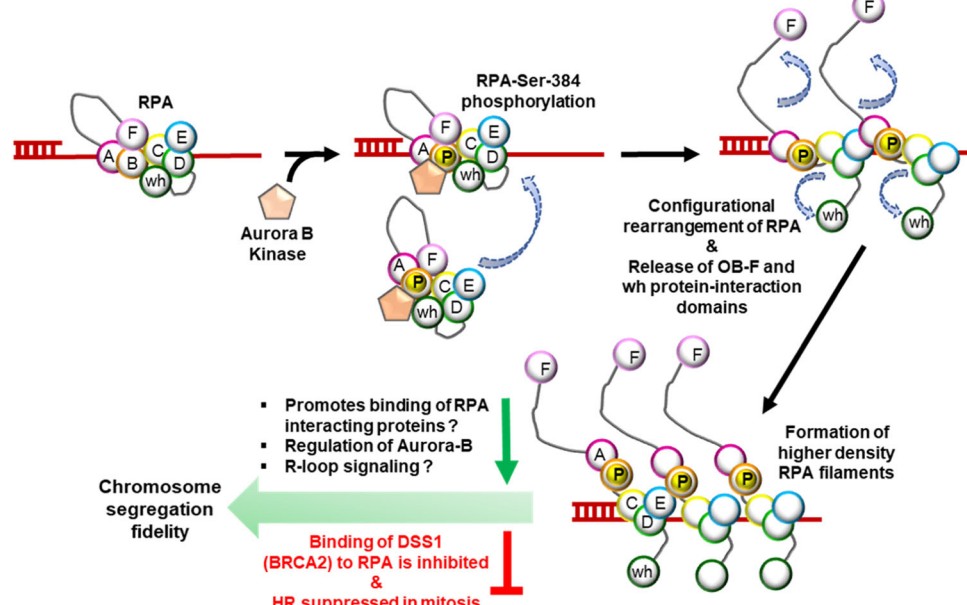

**Fig. 8 | Model for the Aurora B-RPA signaling axis.** During mitosis, RPA bound to ssDNA intermediates or free in solution is phosphorylated by Aurora B kinase at Ser-384 (DBD-B) in the large RPA70 subunit. We propose that phosphorylation releases the protein-interaction domains (OB-F or PID[70N] and wh or PID[32C]) and promotes the formation of higher-density RPA-bound ssDNA filaments. The protein-interaction domains can then promote binding to RPA-interacting proteins.

The feedback mechanism that modulates Aurora B activity through RPA phosphorylation and the involvement of R loops also remains to be elucidated. Since the site of phosphorylation resides close to the DSS1 binding site, recruitment of DSS1-BRCA2 is inhibited leading to suppression of homologous recombination during mitosis. Deregulation of the Aurora B-RPA signaling circuit leads to errors in chromosome segregation fidelity.

RPA binding to DSS1 likely resulting in the inhibition of BRCA2 recruitment and suppression of homologous recombination (HR) in mitosis.

## Discussion

Maintenance of genomic integrity depends on the faithful segregation of newly replicated genomic DNA to the daughter cells during mitosis. How key modulators of DNA metabolism such as RPA are engaged in the complex DNA remodeling processes in mitosis such as chromatin compaction, chromosome segregation, and chromosomal repair has remained poorly understood. Furthermore, how RPA gains functional specificity for specific events in various parts of the cell cycle and tailors ssDNA handoff to other enzymes is also a long-standing question. RPA32 was shown to be phosphorylated in mitosis by CDK1 and phosphorylation promotes mitotic exit in the presence of DNA damage[20]. However, disruption of CDK1-specific mitotic phosphorylation of RPA does not affect normal mitotic progression[69]. Therefore, how RPA activity in mitosis is regulated under normal unstressed conditions and in the absence of DNA damage is not known. Here, we uncover site-specific phosphorylation of RPA by the mitosis-specific Aurora B kinase in the large subunit (RPA70) at Ser-384 within DBD-B. This is the first study showing the cellular relevance of post-translational modifications on RPA70 and emphasizes the importance of following modifications other than in RPA32.

Aurora B kinase is a key component of the chromosomal passenger complex and contributes to numerous processes that maintain the fidelity of chromosome segregation[70]. Aurora B coordinates a complex network of cellular interactions that are tied to its state of activation and localization on the centromeres[71,72]. Interestingly, RPA also localizes to the centromeres. Thus, the spatial regulation of RPA by Aurora B at the centromeres could enable sensing of ssDNA during mitosis and protect ssDNA generated by centromeric stress. Intriguingly, we also uncovered a unique feedback signaling circuit wherein phosphorylation of RPA is important to maintain Aurora B activity and phosphorylation of Aurora B substrates such as Histone H3. Recently,

it was shown that Aurora B activity is regulated through ATR activation at R-loops in the absence of DNA damage and is crucial for chromosome segregation[14]. Effect of ATR on Aurora B activity is mediated through Chk1 without altering the centromeric localization of Aurora B. ATR activation at the centromeres is mediated through RPA recruitment to ssDNA associated with R-loops[14]. Thus RPA-ATR-Chk1 pathway has been proposed to be critical for Aurora B activity. Adding to the complexity of regulation, our results now also position Aurora B upstream of RPA and indicate a feedback circuit wherein phosphorylation of RPA by Aurora B in turn maintains Aurora B activity potentially through R loop signaling mechanisms (Fig. 8). R loops have also been shown to enhance Histone H3 phosphorylation and chromatin compaction[73]. Therefore, it is likely that the effects of RPA phosphorylation on Aurora B activity, S10 phosphorylation of Histone H3, and chromosome compaction reported here are mediated through R loop recruitment of RPA. However, it is likely that RPA directly affects Aurora B activity similar to its direct effect on ATR kinase activity[47].

Additionally, RPA is also a key modulator of other DNA metabolic processes in mitosis such as MiDAS. Future studies will examine the mitotic-specific processes that are regulated by RPA70 phosphorylation such as chromosomal repair, MiDAS, and R-loop resolutions. Aurora B activity is important for faithful chromosome segregation. Thus, disrupting the Aurora B-RPA signaling circuit through loss of RPA phosphorylation leads to defects in chromosome segregation. It is likely that the cells sustaining segregation defects are removed by apoptosis in the subsequent G1 phase through the high basal activation of the p53-mediated checkpoint response. Consistent with these observations, previous studies have shown that delays in mitotic progression can induce p53-dependent cell death response[74,75].

In the crystal structures of RPA bound to DNA, Ser-384 (the site of Aurora B phosphorylation) does not directly contact ssDNA. Through hydrogen-bonding interactions, it positions a neighboring conserved Phe-386 that base-stacks with ssDNA (Fig. 3a). Furthermore, Arg-382 which is a part of the Aurora B recognition motif, interacts with the ssDNA backbone. Thus, upon phosphorylation, this pocket in DBD-B is

likely remodeled. While phosphorylation does not affect overall DNA binding activity, higher density of RPA$^{S384D}$ molecules bind to ssDNA. These findings are further supported by the fact that mutation of Ser-384 to Ala does not influence the DNA binding, configurational states, facilitated exchange, or DSS1 binding properties of RPA (Supplementary Fig. 16). The structural effects are more striking in the overall configurational changes of the domains. PID$^{70N}$ (OB-F) and PID$^{32C}$ (winged-helix domain) are two protein-protein interaction domains situated in RPA70 and RPA32, respectively (Fig. 1a). Conventionally, these PIDs are assumed to be exposed and readily accessible for binding to RPA interacting proteins (RIPs). In yeast RPA, we recently showed that Rtt105 (a chaperone-like protein) binds and configurationally compacts the many domains of RPA, including PID$^{70N}$ and PID$^{32C}$[76]. This phenomenon, termed "configurational stapling", enables Rtt105 to block binding of RIPs (such as Rad52) to RPA. Presence of ssDNA licenses RPA–RIP interactions by remodeling Rtt105. Based on the comparative HDX-MS analysis, we here propose that human RPA has evolved to resemble configurational stapling without the need for a chaperone-regulated mechanism under certain physiological contexts. HDX-MS changes are observed in both PID$^{70N}$ and PID$^{32C}$ when data from RPA and RPA$^{S384D}$ are compared (Fig. 7). These data suggest that these PIDs that are connected to their respective DBDs through flexible linkers are not extended out, but likely in close proximity to the other DBDs. This occlusion likely prevents them from binding to RIPs. Upon phosphorylation by Aurora B, the PIDs are 'licensed' to bind to select RIPs; in this case, promotion of protein interactions with mitosis-specific proteins (Fig. 8). For example, RPA enriched from mitotic cells were shown to exhibit reduced physical interactions with ATM, DNA pol-alpha, and DNA-PK and treatment with phosphatase restored these interactions[29]. The enhancement in intrinsic Trp. fluorescence in the RPA$^{S384D}$ variant further supports a model for release of configurational compaction upon phosphorylation. The compaction and release of RPA could serve as a potential mechanism to regulate RPA-protein interactions in the cell. Such configurations can be further modulated by additional post-translational modifications[77].

RPA–RIP interactions can be positively or negatively regulated by phosphorylation[6,29]. An example of negative regulation is the inhibition of DSS1-BRCA2 binding to the phosphomimetic RPA$^{S384D}$. DSS1-BRCA2 recruitment to ssDNA by RPA is important for resolution of ssDNA intermediates through HR[54]. Intriguingly, HR is suppressed during mitosis[78]. While there are several mechanisms that could contribute to HR suppression in mitosis including the inaccessibility of compacted sister chromatids, we propose that inhibition of BRCA2 recruitment during mitosis provides an additional mechanism for HR suppression in mitosis. DSS1, is a BRCA2 partner, and facilitates the RPA-BRCA2 interaction. The mapped region of interaction between DSS1 and RPA resides around Arg-382 in DBD-B and our findings show that Aurora B phosphorylation at Ser-384 blocks DSS1 binding to RPA that will suppress HR in mitosis. The importance of suppressing BRCA2 activity in mitosis is further highlighted by the phosphorylation of BRCA2 by CDKs at Ser-3291. This modification in BRCA2 blocks physical interaction with RAD51. In the S and G2 phases, phosphorylation at Ser-3291 is low, but drastically increases in mitosis[78]. Thus, RAD51-promoted HR is likely suppressed in mitosis upon phosphorylation of BRCA2. Thus, in addition to CDK regulation of BRCA2, Aurora B phosphorylation of RPA provides added stringency to HR suppression in mitosis. The significance of the DSS1 interaction region is further highlighted by the mutation of the Arg-382 site in certain cancers suggesting that deregulation of this region could underlie genomic instability.

Under conditions where DNA damage is encountered in mitosis, RPA serves as a cellular marker to define anaphase bridges, ultrafine bridges, and as loci-markers for DNA repair events. Recruitment of the Plk1-interacting checkpoint helicase (PICH) chromatin remodeler and the Bloom's syndrome helicase (BLM) helicase limit assembly of histones with centromeric DNA during anaphase[50,79,80]. The activity of both these helicases is required to promote proper resolution of the chromosomes in mitosis. BLM directly interacts with RPA through interactions with PID[70] the binding is required to stimulate unwinding activity of BLM[81]. PICH binds to BLM through its C-terminal region. Interestingly, acidic motifs are found in RPA-interacting proteins that interact with PID$^{70N}$ of RPA, and several such potential motifs are found in PICH. We show here that Aurora B phosphorylation releases PID$^{70N}$ and likely facilitates interactions with RIPs that bind to this domain (Fig. 8). It will be interesting to test whether PICH directly interacts with RPA to function as a complex with BLM during mitosis.

In summary, we propose that Aurora B phosphorylation of RPA acts as a signaling axis that is important for faithful chromosome segregation in mitosis. Thus, future studies of RPA in DNA metabolism should also consider post-translational modifications (especially phosphorylation) at subunits other than RPA32. Tandem modifications at both RPA70 and RPA32 could further yield an additional layer of regulation that might be important in understanding how RPA imparts specificity to DNA repair processes and the maintenance of genomic stability.

## Methods

### Recombinant overproduction and purification of RPA

Human RPA (RPA) was recombinantly expressed using plasmid p11d-hRPA-WT (a kind gift from Marc Wold, Univ. of Iowa). RPA$^{R382Q}$, RPA$^{S384D}$, and RPA$^{S384A}$ mutants were generated in this plasmid background using Q5 site-directed mutagenesis (New England Biolabs). RPA, RPA$^{R382Q}$, RPA$^{S384D}$, and RPA$^{S384A}$ were purified from *E. coli* as described[82] with minor modifications. Briefly, the appropriate plasmids were transformed into Rosetta2(DE3)pLysS cells (Novagen) and transformants on Luria Broth (LB) were selected using ampicillin (100 µg/ml). An overnight culture (10 ml) from a single colony was grown and added to 1 L of LB media containing ampicillin. Cells were grown at 37 °C until the $OD_{600}$ reached 0.6 and then induced with 0.3 mM isopropyl-D-1-thiogalactoside (IPTG). Induction was carried out at 37 °C for 3 h and the harvested cells were resuspended in 120 ml cell resuspension buffer (30 mM HEPES, pH 7.8, 300 mM KCl, 0.02% Tween-20, 1.5X protease inhibitor cocktail, 1 mM PMSF, 10% (v/v) glycerol and 10 mM imidazole). Cells were lysed using 0.4 mg/ml lysozyme followed by sonication. Clarified lysates were fractionated on a Ni$^{2+}$-NTA agarose column (Gold Biotechnology). RPA was eluted using cell resuspension buffer containing 400 mM imidazole. Fractions containing RPA were pooled and diluted with $H_0$ buffer (30 mM HEPES, pH 7.8, 0.02% Tween-20, 1.5X protease inhibitor cocktail, 10% (v/v) glycerol and 0.25 mM EDTA pH 8.0) to match the conductivity of buffer $H_{100}$ ($H_0$ + 100 mM KCl), and further fractionated over a fast-flow Heparin column (Cytiva). RPA was eluted using a linear KCl gradient $H_{100}$–$H_{1500}$, and fractions containing RPA were pooled and concentrated using an Amicon spin concentrator (30 kDa molecular weight cut-off; Millipore Sigma). The concentrated RPA was next loaded onto a HiLoad Superdex S200 column (Cytiva) and fractionated using RPA storage buffer (30 mM HEPES, pH 7.8, 300 mM KCl, 0.25 mM EDTA, 0.02% Tween-20, and 10% (v/v) glycerol). Purified RPA protein was flash-frozen using liquid nitrogen and stored at −70 °C. RPA concentration was measured spectroscopically using $\varepsilon_{280} = 87{,}210$ M$^{-1}$ cm$^{-1}$.

### Generation of fluorescent hRPA

Fluorescently labeled human RPA with Cy3/Cy5 positioned on DBD-A (RPA-DBD-A$^f$) or DBD-D (RPA-DBD-D$^f$) were generated using non-canonical amino acids (ncAA) as described for *S. cerevisiae* RPA[83]. Briefly, 4-azido-*L*-phenylalanine (4AZP) was site-specifically incorporated into DBD-A (RPA70 Ser-215) or DBD-D (RPA32 Trp-107) by engineering TAG stop codons at the corresponding positions in the plasmid using site-directed mutagenesis. A 6x-poly-histidine affinity

tag was also engineered at the C-terminus of RPA70 or RPA32 to isolate RPA-DBD-A[4AZP] or RPA-DBD-D[4AZP] from the truncated non-4AZP incorporated proteins, respectively. The respective plasmid was cotransformed into *E. coli* BL21 Rosetta2(DE3)pLysS cells with a pDuel2-pCNF plasmid that codes for the orthogonal tRNA[UAG] and tRNA synthetase for 4AZP incorporation[32]. Cotransformants were selected using both ampicillin (100 μg/ml) and spectinomycin (50 μg/ml). A 10 ml overnight culture was grown in LB media from a single transformant. The overnight culture was added to 1 L of minimal media optimized for ncAA incorporation[32,83] and grown at 37 °C until the $OD_{600}$ reached 2.0 and then induced with 0.3 mM IPTG. Induction was carried out at 37 °C for 3 h and the harvested cells were resuspended in 120 ml cell resuspension buffer (30 mM HEPES, pH 7.8, 300 mM KCl, 0.02% Tween-20, 1.5X protease inhibitor cocktail, 1 mM PMSF, 10% (v/v) glycerol and 10 mM imidazole). RPA[4AZP] was purified as described above for unlabeled RPA. To fluorescently label the protein, RPA-DBD-A[4AZP] or RPA-DBD-D[4AZP] (~4 μM in 5 ml storage buffer) was mixed with 1.5-fold molar excess DBCO-Cy5 or DBCO-Cy3 (Click Chemistry Tools Inc.) and incubated for 2 h at 4 °C. Labeled RPA was resolved from free dye on a Biogel-P4 column (Bio-Rad Laboratories) using RPA storage buffer. Fluorescent RPA protein was flash-frozen using liquid nitrogen and stored at −70 °C. RPA concentration was measured spectroscopically using $\varepsilon_{280} = 87{,}210 \, M^{-1} \, cm^{-1}$ and labeling efficiency was calculated as described[83].

### Purification of DSS1

A codon-optimized open reading frame for Human DSS1 (DSS1) was synthesized (Genscript Inc.) with a SUMO protease cleavable N-terminal MVKIH-Strep-6x-HIS-SUMO tag. The MVKIH sequence enhances overall protein overproduction[84]. The pRSFDuet-1-MVKIH-STREP-SUMO-DSS1 plasmid was transformed into *E. coli* BL21(DE3) pLysS cells and transformants on LB were selected using kanamycin (50 μg/ml). An overnight culture (10 ml) was grown from a single transformant and added to 1 L of LB media containing kanamycin. Cells were grown at 37 °C until the $OD_{600}$ reached 0.6 and then induced with 1 mM IPTG. Induction was carried out at 37 °C for 3 h and the harvested cells were resuspended in 120 ml cell resuspension buffer (30 mM HEPES, pH 7.8, 300 mM KCl, 0.02% Tween-20, 1.5X protease inhibitor cocktail, 1 mM PMSF, 10% (v/v) glycerol, 1 mM TCEP-HCl, and 10 mM imidazole). Cells were lysed using 0.4 mg/ml lysozyme followed by sonication. Clarified lysates were fractionated on a $Ni^{2+}$-NTA agarose column. DSS1 was eluted using cell resuspension buffer containing 50 mM KCl and 400 mM imidazole. Fractions containing DSS1 were pooled and digested for 18 h at 4 °C with 1:10 molar excess of SUMO protease. The reaction was concentrated using a 30 kDa Amicon spin concentrator and loaded onto a HiLoad Superdex S200 column and fractionated using DSS1 storage buffer (30 mM HEPES, pH 7.8, 300 mM KCl, 0.25 mM EDTA, 0.02% Tween-20, 1 mM TCEP-HCl, and 10% (v/v) glycerol). Purified DSS1 protein was flash-frozen using liquid nitrogen and stored at −70 °C. DSS1 concentration was measured spectroscopically using $\varepsilon_{280} = 17{,}990 \, M^{-1} \, cm^{-1}$.

### In vitro kinase assay

For phosphorylation of RPA and RPA variants (RPA[S384A] and RPA[R382Q]) by Aurora B kinase, recombinant human RPA or RPA-variants (2 μM) were incubated with 250 nM of recombinant Aurora B kinase (Invitrogen Inc.) in kinase reaction buffer (50 mM Tris-HCl, pH 8.0, 10 mM $MgCl_2$, 2 mM TCEP, 10 μM ATP and 20 μCi $^{32}P$-γ-ATP (PerkinElmer Inc.). Reactions (50 μl total) were initiated by adding kinase and reactions were incubated at 30 °C for 30 min. 10 μl of 5X Laemmli buffer was added, boiled for a minute, and 30 μl of the reaction was resolved using 8–12% SDS-PAGE and imaged using Coomassie staining and autoradiography. For analysis of the phosphorylation reaction using mass spectrometry experiments were performed similarly except 500 nM of RPA and variants were used in the reaction along with 50 μM

cold-ATP instead of radiolabeled ATP. Reactions were quenched by adding 10 mM EDTA (final concentration) and by freezing on dry ice and analyzed using MS/MS.

### MS/MS analysis of RPA phosphorylation

In vitro kinase reactions were subjected to TCA/acetone (10% TCA in 50% Acetone final v/v) precipitation for 30 min on ice. Precipitated proteins were spun for 10 min at room temperature (16,000 *g*) and the pellets were washed twice with cold acetone. Protein extracts were re-solubilized and denatured in 15 μl of 8 M Urea in 50 mM $NH_4HCO_3$ (pH 8.5) and subsequently diluted to 60 μl for the reduction step with 2.5 μl of 25 mM DTT and 42.5 μl of 25 mM $NH_4HCO_3$ (pH8.5). The diluted reaction was further incubated at 56 °C for 15 min and cooled on ice to room temperature. 3 μl of 55 mM chloroacetamide was added for alkylation and incubated in darkness at room temperature for 15 min. Reaction was quenched by adding 8 μl of 25 mM DTT. Finally, 6 μl of Trypsin/LysC solution (100 ng/μl 1:1 *Trypsin* (Promega) and *LysC* (FujiFilm)) mix in 25 mM $NH_4HCO_3$] and 23 μl of 25 mM $NH_4HCO_3$ (pH8.5) was added to 100 μl final volume. Digestion was carried out for 2 h at 42 °C and an additional 3 μl of trypsin/LysC mix was then added and digested overnight at 37 °C. The reaction was terminated by acidification with 2.5% trifluoroacetic acid (TFA) to 0.3% final. Digests were desalted using Agilent Bond Elut OMIX C18 SPE pipette tips per manufacturer protocol, eluted in 10 μl of 70/30/0.1% $ACN/H_2O/TFA$, and dried to completion using a speedvac and finally reconstituted in 25 μl of 0.1% formic acid. Peptides were analyzed by nanoLC-MS/MS using an Agilent 1100 nanoflow system (Agilent) connected to hybrid linear ion trap-orbitrap mass spectrometer (LTQ-Orbitrap Elite, Thermo Fisher Scientific) equipped with an EASY-Spray electrospray source (held at constant 35 °C). Chromatography of peptides prior to mass spectral analysis was accomplished using a capillary emitter column (PepMap C18, 3 μM, 100 Å, 150 × 0.075 mm, Thermo Fisher Scientific) onto which 2 μl of extracted peptides was automatically loaded. NanoHPLC system delivered solvents A: 0.1% (v/v) formic acid, and B: 99.9% (v/v) acetonitrile, 0.1% (v/v) formic acid at 0.50 μL/min to load the peptides (over a 30 min period) and 0.3 μl/min to elute peptides directly into the nano-electrospray with gradual gradient from 0% (v/v) B to 30% (v/v) B over 80 min and concluded with 5 min fast gradient from 30% (v/v) B to 50% (v/v) B at which time a 5 min flash-out from 50 to 95% (v/v) B took place. As peptides eluted from the HPLC-column/electrospray source survey MS scans were acquired in the Orbitrap with a resolution of 120,000 followed by CID-type MS/MS fragmentation of 30 most intense peptides detected in the MS1 scan from 350 to 1800 m/z; redundancy was limited by dynamic exclusion.

MS/MS data files were converted to mgf file format using MSConvert (ProteoWizard: Open-Source Software for Rapid Proteomics Tools Development). Resulting mgf files were used to search against Uniprot *Escherichia coli* proteome databases (UP000000625 01/17/2019 download, 4446 total entries) containing user-defined construct sequences along with a cRAP common lab contaminant database (116 total entries) using in-house Mascot search engine 2.2.07 (Matrix Science) with fixed Cysteine carbamidomethylation and variable Serine and Threonine phosphorylation, Methionine oxidation, plus Asparagine or Glutamine deamidation. Peptide mass tolerance was set at 15 ppm and fragment mass at 0.6 Da. Protein annotations, significance of identification, and spectral-based quantification was done with Scaffold software (version 4.11.0, Proteome Software Inc., Portland, OR). Peptide identifications were accepted if they could be established at greater than 96.0% probability to achieve an FDR less than 1.0% by the Scaffold Local FDR algorithm. Protein identifications were accepted if they could be established at greater than 99.0% probability to achieve an FDR less than 1.0% and contained at least 2 identified peptides. Protein probabilities were assigned by the Protein Prophet algorithm[85]. Proteins that contained similar peptides and

could not be differentiated based on MS/MS analysis alone were grouped to satisfy the principles of parsimony. All of the Mascot assigned phophopeptides were subsequently manually interrogated to confirm the identification and residue assignment.

## Cell lines

HCT116 and 293T cells were purchased from American Type Culture Collection. HCT116 cells were maintained in McCoy's 5A (modified) medium supplemented with 10% fetal bovine serum (FBS), and 100 units/ml penicillin and 100 mg/ml streptomycin (Gibco). 293T cells were maintained in Dulbecco's Modified Eagle Medium supplemented with 10% FBS, 100 units/ml penicillin, and 100 mg/ml streptomycin (Gibco).

## CRISPR/Cas9-mediated gene editing

The RPA-S384A lines were created by the Genome Engineering & Stem Cell Center (GESC@MGI) at Washington University in St. Louis. Briefly, synthetic gRNA targeting the sequence (5′- tccaccgaaatcagagactcNGG) and donor single-stranded oligodeoxynucleotides (cttttacaggctgataa atttgatggttctagacagcccgtgttggctatcaaaggagcGcgagtc-Gctgatttcggtgg acggagcctctccgtgctgtcttcaagcactatcattgcgaatcc) used for knock-in were purchased from IDT, complexed with Cas9 recombinant protein and transfected into HCT116 cells. Transfected cells were then single-cell sorted into 96-well plates, and single-cell clones were identified using Next Generation Sequencing to analyze the target site region as those harboring knock-in mutation. Positive clones were expanded, and the genotype confirmed prior to cryopreservation. Two clones were selected for this study. All clones were negative for mycoplasma contamination and authenticated as HCT116 cells by STR profiling.

## Western blotting

Cells were washed twice in phosphate-buffered saline (1X PBS) and lysed in mammalian cell lysis buffer (MCLB) (50 mM Tris-Cl pH 8.0, 5 mM EDTA, 0.5% Igepal, 150 mM NaCl) that was supplemented with the following inhibitors just before lysis: 1 mM phenylmethylsufonyl fluoride (PMSF), 1 mM sodium fluoride, 10 mM β-glycerophosphate, 1 mM sodium vanadate, 2 mM DTT, 1X protease inhibitor cocktail (Sigma–Aldrich), 1X phosphatase inhibitor cocktail (Santa Cruz Biotechnology). Lysates were rocked for 15 min at 4 °C and cleared by centrifugation at 17,968 $g$ for 10 min at 4 °C. Proteins were separated on SDS-PAGE and transferred onto nitrocellulose membranes (0.45 μm; Bio-Rad Laboratories). Membranes were blocked for 1 h in 5% nonfat dry milk dissolved in Tris Buffered Saline with 0.1% Tween 20 (TBS-T). β-Tubulin (1:1000; Cell Signaling, 2128), Vinculin (1:1000; Cell Signaling, 13901), anti-RPA70 (1:1000; Cell Signaling; 2198), p53 (1:1000; Santa Cruz Biotechnology, sc126), Histone H3 (1:2000; Cell Signaling, 14269) and Aurora B (1:1000; Cell Signaling, 3094). All phospho-specific primary antibodies were diluted (1:1000) in 1% nonfat dry milk dissolved in TBS-T buffer and rocked overnight at 4 °C. The following phospho-specific antibodies were used: pS384-RPA70 (monoclonal, custom generated by Genscript), pS10-Histone H3 (Cell Signaling, 53348), pS139-H2AX (Millipore Sigma, 05-636, Cell Signaling, 2577), pS317-Chk1 (Cell Signaling, 12302) and pT232-Aurora B (Rockland, 600-401-677). Membranes were probed with HRP-conjugated secondary antibodies (Jackson ImmunoResearch) diluted 1:30,000 in TBS-T buffer and incubated for 1 h at room temperature except for the secondary antibodies targeted against pS384-RPA70 that were diluted in 1–2% nonfat dry milk dissolved in TBS-T buffer. Membranes were developed using ECL substrate (Pierce) and chemiluminescence was captured using iBright CL1500 imager (Thermo Fisher). Blots were quantitated using the iBright analysis software (Thermo Fisher). To assess specificity of pS384-RPA70 antibody, cold kinase reactions were set up as indicated above and reactions were probed with the indicated antibodies.

## Phosphatase treatment

For phosphatase treatment, HCT116 cell lysates were collected in MCLB buffer without EDTA and phosphatase inhibitors. 70 μg of total protein was added to 30 μL of calf intestinal phosphatase (CIP) reaction buffer (100 mM NaCl, 50 mM Tris pH 8.0, 10 mM MgCl₂, 1 mM DTT and 0.7X EDTA free-protease inhibitor cocktail). Diluted lysates were treated with either 14 μL of quick CIP (NEB) or CIP storage buffer (25 mM Tris pH 7.6, 1 mM MgCl₂, and 70% glycerol v/v). Reactions were then incubated at 37 °C for 1 h followed by boiling in 1X Laemmli buffer and probed by western blotting.

## Knockdown and rescue of Aurora B

To knockdown Aurora B, pLKO.1 vectors expressing shRNA (TRCN0000000776, Sigma) targeting the 3′UTR of Aurora B was used and shRNA targeting luciferase (Sigma) was used as control. To rescue Aurora B expression, pcDNA3.1 plasmid carrying myc-tagged (N-terminus) Aurora B was generated (Genscript). For transfections, $1.8 \times 10^6$ HCT116 cells were plated per 60-mm dish and transfected with 4.5 μg of pLKO.1 plasmid expressing shRNA using 20 μl of Lipofectamine 2000 (Invitrogen). For rescue experiments, cells were also transfected with 2 μg of either pcDNA3.1-myc-AurKB or pCDNA3.1 empty vector. 30 h later, culture media was replaced with fresh media supplemented with 75 ng/mL of nocodazole (Sigma-Aldrich) and incubated for 18 h. Mitotic cells were collected by shake-off method.

## Cell viability assays

5000 HCT116 cells expressing RPA-WT or RPA-S384A were cultured per well of a 96-well plate in McCoy's culture medium without phenol red. 20 μL of MTS solution (CellTiter 96 Aqueous One solution reagent, Promega) was added to each well, and the plates were incubated for 1 h. Absorbances were read at 490 nm (Synergy H1, BioTek). To obtain background-corrected absorbances, average absorbance values of media with MTS only were subtracted from all other absorbances. To determine the rate of cell proliferation, final absorbances were normalized to 0 h absorbance of the respective cells. Apoptosis was measured using the Caspase-Glo® 3/7 assay system (Promega). Cells were seeded at a density of $1 \times 10^4$ in 96-well black polystyrene microplates (Corning). 100 μL of caspase-Glo reagent, including caspase-Glo substrate, and caspase-Glo buffer was added to each well and incubated for 90 min at RT in the dark. Luminescence was measured using a multi-mode microplate reader (Synergy H1, BioTek). To induce replication stress, cells were treated with 10 ng/μL of SN-38 dissolved in DMSO (Tocris). To obtain background-corrected absorbances, average absorbance values of media with reagent only were subtracted from all other absorbances.

## Cell cycle analysis

To synchronize HCT116 cells in mitosis (prometaphase), $1.5 \times 10^6$ cells were seeded per 100 mm dish. Next day, cells were treated with 75 ng/mL nocodazole (Sigma-Aldrich) and incubated for 18 h. Rounded mitotic cells were collected by shake-off method and centrifuged at 931 $g$ for 4 min at 4 °C. Cells were washed twice with cold PBS. The cell pellets were lysed in MCLB buffer as described above and collected for western blot analysis. To release cells from mitotic arrest into G1 phase, mitotic cells were collected by shake-off and washed twice in PBS and resuspended in culture media. Cells were then plated onto 100 mm dishes and incubated for 3 h. Control cells were treated with vehicle-0.1% DMSO. For inhibition of Aurora kinase B, cells were arrested in mitosis as described above and treated with 3 μM Aurora kinase B inhibitor (AZD1152) or 0.1% DMSO (vehicle) for an additional 45 min. Rounded mitotic cells were collected as described above and centrifuged at 931 $g$ for 2–4 min at 4 °C. Cells were washed twice with cold 1XPBS and lysed in MCLB buffer. For cell synchronization using double thymidine block, $1.2 \times 10^6$ HCT116 cells were seeded per 100 mm dish, and next day, cells were treated with

2 mM thymidine (Sigma-Aldrich) for 16 h. Cells were then washed twice with 1XPBS and cultured in media without inhibitors for 8 h. For the second thymidine block, cells were cultured in media supplemented with 2 mM thymidine for 16 h. Cells were released from G1/S phase arrest by washing twice with 1X PBS and cultured in media without inhibitors. Cells were collected at 3 h and 6 h post-release from G1/S phase arrest.

## Immunofluorescence

HCT116 cells were grown on poly-d-lysine–coated glass coverslips (Neuvitro) in a 12-well cell culture plate. To arrest cells in mitosis, cells were treated with either 0.1% DMSO (vehicle) or 75 ng/ml nocodazole (Sigma-Aldrich) for 18 h. To release cells from mitotic arrest, rounded cells were collected by mitotic shake-off and centrifuged at 931 $g$ for 3 min at RT. Cells were washed twice with PBS with gentle inversion and resuspended in culture media. Cells released from mitotic arrest were grown on poly-d-lysine–coated glass coverslips (Neuvitro) for 1 h at 37 °C. Cells were fixed in 4% paraformaldehyde overnight at 4 °C. For immunostaining, cells were permeabilized with 2% Triton X-100 diluted in 1X PBS for 15 min and incubated in blocking buffer (2% BSA, 0.1% Igepal, 1× PBS) for 30 min. The following primary antibodies diluted in blocking buffer were used: α-Tubulin (1:45; Cell Signaling, 2125), pS10-Histone H3 (1:1000; Millipore Sigma, 05-1336). Cells were incubated with antibodies in a humidified chamber at RT for 1 h. Coverslips were rinsed in wash buffer (0.1% Igepal dissolved in 1X PBS) three times and the following secondary antibodies diluted in blocking buffer were used: Goat anti-rabbit Alexa Fluor 488 (1:500; Thermo Fisher), Cy3-conjugated AffiniPure donkey anti-mouse (1:200, Jackson ImmunoResearch). Cells were incubated with the secondary antibodies in a humidified chamber at RT for 45 min and washed four times in a wash buffer. Coverslips were mounted onto frosted microscope slides (Fisher) using ProLong Gold antifade reagent with DAPI (Invitrogen). Immunostained cells were analyzed under Leica DM6 B upright fluorescent microscope with a 100× oil immersion objective. The digital images were acquired using Leica DFC 9000GT camera and processed by LAS X imaging software. LUT is linear and covers the full data range. Greater than 70 cells in anaphase/telophase were counted to calculate the frequency of lagging chromosomes and anaphase bridges. For surface area measurements of pS10 Histone H3 stained nuclei, greater than 200 nuclei from three independent replicates were analyzed using ImageJ software (NIH).

## Flow cytometry

Adherent cells were trypsinized and resuspended in culture media. Cells were centrifuged at 274 $g$ for 2 min at 4 °C. Mitotic cells were collected by shake-off method as described above. Cell pellets were washed once with cold 1× PBS. Cells were then gently resuspended in 500 μL of cold PBS. The cell suspension was added to 5 mL of 100% ethanol dropwise with mild vortexing and stored at −20 °C. To stain with propidium iodide (PI), fixed cells were centrifuged at 931 $g$ for 2 min, and pellets were resuspended in 1% bovine serum albumin (BSA) diluted in 1× PBS. Cells were counted and resuspended in PI solution containing 3/50 volume of 50× PI (Sigma-Aldrich), 1/40 volume of 10 mg/mL RNAaseA (Thermo Scientific) and 1% BSA diluted in 1× PBS. Samples were filtered through 35 um strainer caps (Corning) and incubated in the dark for 30 min at RT. Samples were analyzed on a BD FACSCanto II flow cytometer using BD FACSDiva software (BD Biosciences). Samples were collected using a low flow rate, and a minimum of 15,000 cycling events were recorded. Data analyses were performed using ModFit LT software (Verity Software).

## Measurement of DNA binding kinetics using stopped-flow fluorescence

All stopped-flow experiments were performed on a SX20 instrument (Applied Photophysics Inc.) at 25 °C in RPA reaction buffer (30 mM HEPES pH 7.8, 100 mM KCl, 5 mM MgCl$_2$, 6% (v/v) glycerol, and 1 mM b-ME). The respective mixing schemes are denoted by cartoons alongside the respective figure panels. Seven to eight individual shots were averaged for each experiment. All experiments were repeated a minimum of $n = 3$ times and the mean value and SEM calculated from the individual fits are reported. To capture RPA binding to ssDNA using intrinsic Trp fluorescence, RPA (100 nM) from one syringe was mixed with increasing concentrations of (dT)$_{35}$ ssDNA (0–800 nM) from the other syringe and the change in Trp fluorescence was measured as a function of time by exciting the sample at 290 nm and collecting fluorescence emission using a 305 nm long-pass filter. Data were fit to a single exponential equation and a plot of the $k_{obs}$ (s$^{-1}$) as a function of [RPA] yielded $k_{on}$ and $k_{off}$ values.

A Förster resonance energy transfer (FRET) based experiment was developed to investigate the facilitated exchange activity of RPA. Here, assembly of multiple fluorescent RPA molecules on a single ssDNA substrate is captured. RPA molecules are labeled on either DBD-A with Cy5 or DBD-D with Cy3. When these molecules are situated adjacent to each other, an increase in FRET is observed. To measure facilitated exchange, these high-FRET fluorescent RPA-containing filaments were challenged with unlabeled RPA and the change in FRET was measured. 200 nM RPA-DBD-A$^{Cy5}$ and 200 nM RPA-DBD-D$^{Cy3}$ were premixed with 90 nM (dT)$_{97}$ and shot against unlabeled RPA (500 nM). Samples were excited at 535 nM (Cy3 wavelength) and Cy5 emission was captured using a 645 nm long-pass filter. Data were fit to s single exponential plus linear equation.

## Hydrogen-Deuterium Exchange mass spectrometry (HDX-MS) analysis

Stock solutions of RPA (13.4 mg/mL) were mixed in the presence or absence of (dT)$_{35}$ ssDNA in a 1:1.2 ratio. Reactions were diluted 1:10 into deuterated reaction buffer (30 mM HEPES, 200 mM KCl, pH 7.8) at 22 °C. Control samples were diluted into a non-deuterated reaction buffer. At each time point (0, 0.008, 0.05, 0.5, 3, 30 h), 10 μL of the reaction was removed and quenched by adding 60 μL of 0.75% formic acid (FA, Sigma) and 0.25 mg/mL porcine pepsin (Sigma) at pH 2.5 on ice. Each sample was digested for 2 min with vortexing every 30 s and flash-frozen in liquid nitrogen. Samples were stored in liquid nitrogen until the LC-MS analysis. LC-MS analysis of RPA was completed as described[46]. Briefly, the LC-MS analysis of RPA was completed on a 1290 UPLC series chromatography stack (Agilent Technologies) coupled with a 6538 UHD Accurate-Mass QTOF LC/MS mass spectrometer (Agilent Technologies). Peptides were separated on a reverse phase column (Phenomenex Onyx Monolithic C18 column, 100 ×2 mm) at 1 °C using a flow rate of 500 μl/min under the following conditions: 1.0 min, 5% B; 1.0 to 9.0 min, 5 to 45% B; 9.0 to 11.8 min, 45 to 95% B; 11.8 to 12.0 min, 5% B; solvent A = 0.1% FA (Sigma) in water (Thermo Fisher) and solvent B = 0.1% FA in acetonitrile (Thermo Fisher). Data were acquired at 2 Hz s$^{-1}$ over the scan range 50–1700 m/z in the positive mode. Electrospray settings were as follows: the nebulizer set to 3.7 bar, drying gas at 8.0 L/min, drying temperature at 350 °C, and capillary voltage at 3.5 kV. Peptides were identified as previously described[86–89] using MassHunter Qualitative Analysis, version 6.0 (Agilent Technologies), Peptide Analysis Worksheet (ProteoMetrics LLC), and PeptideShaker, version 1.16.42, paired with SearchGUI, version 3.3.16 (CompOmics). Deuterium uptake was determined and manually confirmed using HDExaminer, version 2.5.1 (Sierra Analytics). Heat maps were created using MSTools[90].

## RPA-ssDNA binding was measured using fluorescence anisotropy

5′-FAM-(dT)$_{20}$ or 5′-FAM-(dT)$_{40}$ were diluted to 10 nM in DNA binding buffer (50 mM Tris-acetate pH 7.5, 50 mM KCl, 5 mM MgCl$_2$, 1 mM DTT, 10% (v/v) glycerol, and 0.2 mg/ml BSA). 1.2 ml of this ssDNA stock was taken in 10 mm pathlength quartz cuvettes (Firefly Inc.)

maintained at 23 °C and G-factor corrected fluorescence anisotropy was measured (in triplicate) using a PC1 spectrofluorometer (ISS Inc.). Data acquisition was performed using the instrument-associate Vinci 3 software. Samples were excited at 488 nm and the resulting fluorescence emission was collected using a 520 nm band-pass emission filter. To extract corrected anisotropy from raw values (instrument readings) the concentrations of ssDNA, total fluorescence, and the added protein concentration were first corrected for the effect of sample dilution resulting from the stepwise addition of the protein. Second, the FAM anisotropy was corrected for changes in the fluorescence quantum yield of the bound species due to proximity of the fluorescein moiety to the protein (often referred to as protein-induced fluorescence enhancement) in order to plot the anisotropy, change due to reduction in rotational correlation time alone i.e., increase in molecular weight of the ssDNA due to complex formation. For this, the dilution-corrected FAM fluorescence values were used to correct and rescale anisotropy values as described[76]. The saturation points were taken as the intersection of biphasic or triphasic curves from the linear fits of the initial data points reflecting the change in anisotropy upon binding of sub-saturating amounts of proteins.

### Circular dichroism measurements

CD measurements were performed using a Chirascan V100 spectrometer (Applied Photophysics Inc.). A nitrogen-fused setup with a cell path of 1 mm was used to perform the experiments at 20 °C. All CD traces were obtained between 200 and 260 nm, and traces were background corrected using CD reaction buffer (100 mM NaF, 1 mM TCEP-HCl, and 5 mM Tris-HCl pH 7.5). 600 nM of RPA, RPA$^{R382Q}$, RPA$^{S384A}$, or RPA$^{S384D}$ was used, and 10 scans were collected and averaged per sample using 1 nm step size and 1 nm bandwidth.

### Bio-layer interferometry (BLI) to capture DSS1-RPA interactions

BLI experiments were performed using a single-channel BLItz instrument (Sartorius) in advanced kinetics mode at 25 °C with shaking at 2200 rpm. For protein binding, streptavidin (SA) biosensors were pre-hydrated by incubating the tips in BLI buffer (30 mM HEPES pH 7.8, 100 mM KCl, 5 mM MgCl$_2$, and 6% (v/v) glycerol) for 10 min. The experiment was performed as sequential steps in BLI buffer: i) initial baseline was recorded for 30 s, ii) the tip was incubated with 300 nM of Strep-tagged DSS1 for 120 s, iii) tip was washed with buffer for 30 s, iv) binding to different concentrations of RPA was assessed for 120 s, and v) dissociation was recorded with buffer for 120 s. Sensorgrams were normalized to the buffer signal. All experiments were repeated at least three times.

### Size exclusion chromatography

SEC experiments to capture the formation of RPA nucleoprotein filaments on ssDNA were performed using 600 μl of 3 μM RPA in the absence or presence of ssDNA (3 μM (dT)$_{35}$ or 1 μM (dT)$_{97}$). RPA and ssDNA were premixed and incubated for 10 min on ice and resolved using a Superose 6 Increase (10/300) size exclusion column (Cytiva) equilibrated with SEC buffer (30 mM HEPES, pH 7.8, 100 mM KCl, 1 mM TCEP-HCl, and 10% (v/v) glycerol). 0.5 ml fractions were collected and assessed using SDS-PAGE.

### Reporting summary

Further information on research design is available in the Nature Portfolio Reporting Summary linked to this article.

## Data availability

All data are contained within the manuscript. Plasmids used for protein overexpression are available upon request. Source data are provided as a Source Data file. Source data are provided with this paper.

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

## Acknowledgements

We thank Dr. Grzegorz Sabat (University of Wisconsin Madison-Mass Spectrometry Core), and GESC@MGI at Washington University in St. Louis for their technical support. We thank Dr. Jaigeeth Deveryshetty for his assistance with generating the structural model of RPA. This work was supported by grants from the NIGMS, National Institutes of Health R01 GM143179 (S.O.), R01 GM130746, R01 GM133967, and R35 GM149320 (E.A.). AUC experiments were supported by a grant from the Office of the Director, National Institutes of Health, S10 OD030343 (E.A.). Funding for the Proteomics, Metabolomics and Mass Spectrometry Facility at MSU was made possible in part by the MJ Murdock Charitable Trust and NIGMS of the National Institutes of Health P20 GM103474 and S10 OD28650 (B.B.). S.K. is supported by a grant from the National Cancer Institute F99 CA274696.

## Author contributions

P.R., S.K., and J.R.M. performed and analyzed a major part of the experiments. V.K., R.C., N.P., B.R.T., and A.B., helped perform a minor portion of the experiments and analysis. S.O., E.A., and B.B. devised the project and the experiments. S.O., E.A., and B.B. wrote the manuscript with assistance and contributions from all authors.

## Competing interests

The authors declare no competing interests.
