## [Peer Review File · Nature Communications]

An Aurora B-RPA signaling axis secures chromosome segregation fidelityREVIEWER COMMENTS

Reviewer #1 (Remarks to the Author):

The manuscript by Poonam and colleagues identifies a function for Aurora B kinase to regulate the function of RPA. First, it is demonstrated that there is an Aurora B motif in the OB fold-B domain of RPA1. Using a phospho-specific antibody, it is quite convincingly demonstrated that Aurora B targets this residue in mitosis, and the modification is specific to S384 of RPA1. Next, using a variety of elegant biochemical assays, it is shown that the phosphomimetic mutant (S384D) has somewhat altered ssDNA binding properties: its affinity to ssDNA is slightly lower, yet at the same time its binding is less dynamic than that of the wild type protein. Additionally, the S384D mutant protein exhibits reduced interaction with DSS1, a partner of BRCA2. Subsequently, using multiple equally well-performed cellular assays, it is shown that the non-phosphorylatable mutant (S384A) reduces cell viability, exhibits sensitivity to replication stress, and shows chromosomal segregation defects. With one important caveat (see below), I am very supportive of the manuscript.

Specific comments:

Main point:

The study is based on the presumptions that the SD mutation is phosphomimetic (1), while the SA mutant represents RPA that cannot be phosphorylated (2).

I generally agree with the point (1), as there is likely nothing else that can be done. Have the authors observed altered properties of RPA upon in vitro phosphorylation by Aurora B? I suppose the efficacy of in vitro phosphorylation is too low, which necessitates the use of the mutant.

However, there can be a problem with point (2). Because the S384 residue is localized in the high affinity DNA binding domain, the mutation into alanine per se might affect its function, instead of only preventing phosphorylation by Aurora B. Consequently, the observed cellular effects may not be linked to the missing phosphorylation. Additionally, as the phosphorylation site is close to the interaction motif with DSS1, a similar concern also applies here. Therefore, I feel that an essential control is to analyze biochemically the S384A mutant. In case the S384A mutant does not behave as wild type, additional evidence linking Aurora B modification to the altered RPA mutant phenotype would be necessary.

Additional minor points

- RPA-ssDNA complexes triggers the ATM/ATR cellular DNA damage checkpoint response  this is too much simplified, ATM will stop being activated once DNA breaks get resected
- page 6: "Aurora B kinase only phosphorylates"  "primarily phosphorylates"
- Figure 2: please include a phosphatase control when introducing the phospho-specific antibody
- The sentence at the top of page 9 is confusing, first part compares mutant to wild type, and the second part wild type to mutant. Please unify.

Reviewer #2 (Remarks to the Author):

In this study, the authors identified and validated a Serine residue (384) on RPA70 that is phosphorylated by Aurora B kinase during mitosis. They conducted a series of complementary biochemical and cellular experiments to show that phosphorylation at this residue promotes cell viability and recovery from replication stress, higher density RPA filaments and conformational changes that impair binding to DSS1. The authors include a model that suggests phosphorylation of RPA70 likely suppresses homologous recombination during mitosis. The manuscript is very well written, and the study includes critical controls which increase the rigor. Given the multiple roles of

RPA in DNA replication and repair, the identification of a novel regulatory site on RPA70 during mitosis should be of broad interest. A few comments that require clarification are below.

1. I commend the authors for including the critical control of homozygous knock-in Ser-384-Ala to validate the specificity of their custom-made pS384-RPA70. However, validation of the Ser-384-Ala genotype should also be included. Also, is the 2nd clone (supplementary Fig 5) also a homozygous knock in? Looks like there is a band with the pS384-RPA708 antibody for this clone (Supplementary Figure 5).

2. Page 8, paragraph 1. In the last sentence, it is not clear what the authors mean by "genomic stress". It is possible that Ser-384 is induced by other DNA damaging agents, rather than just a reagent that induces replication stress?

3. If RPA is more efficient at facilitated exchange (FE) than RPA-S384D, wouldn't that make the unphosphorylated form easier to remodel? I assume this depends on how FE is defined. The last sentence of paragraph 1, page 11 states phosphorylation enables easier remodeling. Please clarify.

4. Supplementary Fig 6. Quantification of the p53 and gammaH2AX levels in the WT and Ser-384-Ala cells would strengthen the results and rigor. The higher basal level of p53 expression in the mutant is very interesting.

5. Fig. 6, did the authors observe an increase in micronuclei in the Ser-384-Ala mutants? This would be expected given the increase in anaphase bridges, which could lead to breakage.

6. Background information in the Discussion could be reduced. However, this section should also include a discussion of why the authors think the Ser-384-Ala cells show more apoptosis, higher basal levels of p53, and greater sensitivity to replication stress than wild type cells.

7. Minor comment: page 18, "tailor's" is misspelled.

Reviewer #3 (Remarks to the Author):

An Aurora B-RPA signaling axis secures chromosome segregation fidelity

In this manuscript, the authors report the regulation of RPA by the mitotic kinase AURKB. The authors show that RPA70 can be phosphorylated in vitro by AURKB and only at S384 which is located in a consensus sequence of AURKB.

S384-RPA70 was also found to be phosphorylated in vivo in several phosphoproteomic studies. In vivo, phosphorylation of S384-RPA70 occurs only in mitosis and follows phosphorylation of S10-H3.

In the manuscript all data were obtained by analyzing in vivo and in vitro, the behavior of the phosphomimetic mutant S384D or the mutant S384A that cannot be phosphorylated.

Ser-384 is located on the DNA binding domain of the protein but the S383D mutant of RPA70 shows the same ssDNA binding properties of RPA as the wild type protein, only the dynamics of the binding seems to be different. However, the S383D mutant induces a higher density of RPA molecules bound to ssDNA.

The S384A mutant affects cell viability with hypersensitivity to DNA damage. The S384D mutant inhibits homologous recombination by preventing recruitment of DSS1-BRCA2.

I have two major comments

1 - The involvement of AURKB in S384 phosphorylation in vivo was evaluated here using the anti-AURKB inhibitor AZD1152. The use of inhibitors without rescue is always questionable, and the entire manuscript is based on this result.

A cleaner approach is needed. A depletion of AURKB by siRNA and a rescue with an AUKB resistant to siRNA is currently used in the field (or the use of an inhibitor and a rescue with a kinase

resistant to the inhibitor). This would unambiguously prove the involvement of AURKB in S384-RPA70 phosphorylation in vivo and will strengthen the data. Only when this is demonstrated can the authors say or conclude that phosphorylation of RPA70 by AURKB induces ...

This is mandatory.

Indeed, many sentences (or titles), such as the following, border on over-interpretation

Few examples: (there are many)

Page 6 title

"RPA phosphorylation by Aurora B is specific to mitosis."

should be

"Phosphorylation of RPA70 at S384 is specific to mitosis"

Page 10

"Thus, we propose that Aurora B phosphorylation of RPA is changing the arrangement of DBD-B and this configurational change promotes higher density of RPA molecules bound on ssDNA ."

should be:

"Thus, we propose that phosphorylation of RPA70 at S384 is changing the arrangement of DBD-B and this configurational change promotes higher density of RPA molecules bound on ssDNA".

Page 10 title

"Aurora B phosphorylation promotes formation of higher density RPA binding to ssDNA".

Should be

"Phosphorylation of RPA70 at S384 promotes formation of higher density RPA binding to ssDNA"

2 - Mitotic phosphorylation of RPA70 is critical for chromosome segregation.

« Intriguingly, when we probed for Ser-10 phosphorylation of Histone H3 (the mitotic marker), we observed a 50% decrease in Ser-10 phosphorylation in the phospho- dead mutant (Figures 6a & b) ».

This is indeed intriguing because 100% Ser10-H3 phosphorylation marks mitosis, a decrease to 50% is a surprise since it is induced only by the presence of S384A-RPA70 or the absence of phosphorytable S384A-RPA70.

How this can affect phosphorylation of Ser10-H3?

Does it affect AURKB activity? localization? ... Because the phenotype observed is reminiscent to an AURKB defect.

For instance, what about the following hypothesis:

Would it be possible that Ser-384 must be phosphorylated in mitosis to avoid DDR ?

If Ser-384 is not phosphorylated, DDR is activated and induces an inhibition of AURKB, then a decrease in Ser10-H3 phosphorylation and classic mitotic defects due to a lack of AURKB activity?

« Thus, these chromosomal defects clearly indicate that Ser-384 phosphorylation of RPA70 is important for segregation of chromosomes in mitosis ».

Well, the authors should investigate/provide a mechanism to explain this because it might be a side effect. I mean Ser-384 phosphorylation of RPA70 might indirectly affect segregation (see above).

Reviewer #4 (Remarks to the Author):

In this study, Roshan et al present cellular, biophysical, and biochemical investigations into how phosphorylation (at Ser384) can remodel the protein interaction domains of Aurora B-RPA and how, in turn, this can influence its binding to double-stranded DNA. A broad range of assays and techniques were used to support their conclusions, which seemed to have well-constructed hypotheses based on previous findings in the field and their lab. The application of HDX-MS offers important information on the conformational dynamics of the system and how it is affected by DNA

binding and mutation/phosphorylation, and adds value and scope to their conclusions on the structural biology of Aurora B-RPA.

I am not an expert on kinases or chromosome segregation so am ill equipped to judge the novelty of the findings and their importance to the field. I am however well equipped to assess the mass spectrometry experiments and have some major concerns with the level of detail provided on the HDX-MS methodology, results, and explanation of the technique which the authors should address below.

Major concerns:

In the main text (p. 15 onwards) the authors describe that configurational changes bring about uptake or loss of deuterium. This is a misunderstanding of the HDX-MS experiment performed here. Loss should not occur during HDX in relation to the conformational dynamics of your protein system (as you are in high D₂O concentration buffer - >90% D₂O). What is presented here is a differential HDX experiment, where you have the HDX of one state minus the HDX of another state, where the conditions are identical except for the difference in state (which can be a mutation, ligand binding, etc). This needs to be described in the correct way in the text, indeed the authors have used the correct phrasing the figure legend of Fig. 7 (more uptake and less uptake in state X (in this case the mutation)).

There is a lack of required detail in the methods and supporting information for a reader to assess the quality of the HDX-MS experiments and how they can themselves reproduce them. I would recommend that the authors read the 'White paper' on reporting HDX-MS experiments, found here: <https://www.nature.com/articles/s41592-019-0459-y>

There is a requirement by the community that HDX-MS data meet minimum requirements in their reporting – particularly the requirement for the reporting of:

- Dataset Protein state (one column for each condition: that is, apo, ligand-bound, mutant and others)
- HDX reaction details Labeling conditions, for example, percent D₂O, pH(read), temperature and so on
- HDX time course Listing of what time points were analyzed
- HDX controls Description of HDX control samples analyzed
- Back-exchange Back-exchange (average) for all peptides measured (model system or studied protein) and the interquartile range of these values
- Number of peptides Description of the number of peptides used for which HDX data were obtained
- Sequence coverage Expressed as the percentage of amides covered by the peptides for which HDX data were obtained
- Average peptide length/redundancy Average peptide length and number of readings for any amide (calculated as the total number of peptides for which HDX data were obtained over the total number of amides)
- Replicates (biological or technical) Number and specification of replicate HDX-MS measurements performed for each condition and deuterium incorporation time point
- Repeatability A quantitative measure of the repeatability of deuterium measurement (for example, the average standard deviation from technical replicate measurements of the deuterium content of all peptides from one or more time points for a single condition)
- Significant differences in HDX A value used as a threshold to represent a significant difference in HDX between examined protein states as based on a quantitative measure of repeatability

Many of these are missing from the manuscript currently.

Another important aspect that needs to be discussed is the %protein that is predicted to be bound to its ligand (in this case DNA) during those HDX experiments. As the amount of complex in your sample greatly effects the extent (and therefore the level of signal-to-noise) and type of HDX observed. Is the 1:1.2 protein:DNA ratio chosen enough to achieve sufficient complex formation during the entire HDX experiment – especially, as this sample is then diluted 1:10 in deuterated

buffer (with no ligand present) which would change the equilibrium during the HDX measurements. Consideration of what populations you have in your HDX solution (free protein and protein in complex) can have a drastic influence on the HDX data and, importantly, the conclusions you can make on the data.

Another aspect to consider is oligomerisation: You have shown that DNA binding leads to different oligomer formation, with difference observed between WT and phosphomimetic mutant S384D. However, in the main text, you discuss that the S384D would release the RPA's various domains from its configurationally compacted form to promote RPA-protein interactions through these domains. Could you expand the discussion here to consider what effect the oligomerisation observed could have on the HDX-MS data observed. Does it lead to any difference in HDX kinetics – e.g. EX1 or EXX observed in the raw MS spectra, or is all your data EX2 kinetics?

Additionally, what is the differential HDX for the WT or S384D vs. WT or S384D + DNA? This would help to untangle whether the HDX effects observed are due to DNA binding or the mutant solely.

In your uptake plots for the 13-25 and 166-190 peptides you observe losses of deuterium, either loss of large deuteration in the earliest time points for 13-25 peptide or varied losses and gains of deuterium for 166-190 peptide. This HDX uptake is irregular to observe for meaningful HDX data, which typically increase – or stays the same level – over HDX time. Additionally, the error bars for 13-25 peptide are also very large for what is expected for technical replicates of HDX data. How do you explain these results? They could be due to experimental factors, for instance carry-over and/or back exchange, or possibly due to changes in the equilibrium of oligomerisation or ligand unbinding over time. Addressing the experimental details of complex and oligomer formation predicted discussed above (and reporting the raw MS spectra for this data) will help to understand whether these results are real or artefactual.

Response to Reviewer Comments

We thank the four reviewers for their support of our work and for the thoughtful comments. Our responses are denoted in blue.

Reviewer#1

The manuscript by Poonam and colleagues identifies a function for Aurora B kinase to regulate the function of RPA. First, it is demonstrated that there is an Aurora B motif in the OB fold-B domain of RPA1. Using a phospho-specific antibody, it is quite convincingly demonstrated that Aurora B targets this residue in mitosis, and the modification is specific to S384 of RPA1. Next, using a variety of elegant biochemical assays, it is shown that the phosphomimetic mutant (S384D) has somewhat altered ssDNA binding properties: its affinity to ssDNA is slightly lower, yet at the same time its binding is less dynamic than that of the wild type protein. Additionally, the S384D mutant protein exhibits reduced interaction with DSS1, a partner of BRCA2. Subsequently, using multiple equally well-performed cellular assays, it is shown that the non-phosphorylatable mutant (S384A) reduces cell viability, exhibits sensitivity to replication stress, and shows chromosomal segregation defects. With one important caveat (see below), I am very supportive of the manuscript.

Main point: The study is based on the presumptions that the SD mutation is phosphomimetic (1), while the SA mutant represents RPA that cannot be phosphorylated (2). I generally agree with the point (1), as there is likely nothing else that can be done. Have the authors observed altered properties of RPA upon in vitro phosphorylation by Aurora B? I suppose the efficacy of in vitro phosphorylation is too low, which necessitates the use of the mutant. However, there can be a problem with point (2). Because the S384 residue is localized in the high affinity DNA binding domain, the mutation into alanine per se might affect its function, instead of only preventing phosphorylation by Aurora B. Consequently, the observed cellular effects may not be linked to the missing phosphorylation. Additionally, as the phosphorylation site is close to the interaction motif with DSS1, a similar concern also applies here. Therefore, I feel that an essential control is to analyze biochemically the S384A mutant. In case the S384A mutant does not behave as wild type, additional evidence linking Aurora B modification to the altered RPA mutant phenotype would be necessary.

Supplementary Figure 16. RPA carrying a Ser-384 to Ala substitution behaves similar to wildtype RPA. RPA and RPA^{S384A} were compared for various biochemical activities and show very similar activities. **A)** Similar intrinsic Trp fluorescence profiles and quenching upon ssDNA binding. **B)** Secondary structures as measured by circular dichroism were also similar. ssDNA binding to **C)** short or **D)** longer ssDNA were also similar. **E)** In stopped flow facilitated exchange experiments both proteins remodeled fluorescent RPA bound to ssDNA with similar kinetics and efficiency. **F)** Finally, both proteins bound with similar affinity profiles to DSS1 in biolayer interferometry analysis.

With respect to the S384A comment, as suggested by the referee, we have performed several experiments with the RPA^{S384A} variant and find that it behaves similar to the wild-type protein. The intrinsic Trp signal and quenching by ssDNA are similar (Sup. Fig.16A). The CD spectra shows no difference in secondary structure between RPA and RPA^{S384A} (Sup. Fig.16B). DNA binding is stoichiometric for both proteins on short and long ssDNA substrates (Sup. Fig.16 C & D). In facilitated exchange stop flow analysis, there is no observable difference between RPA and RPA^{S384A} (Sup. Fig.16E). Finally, we also tested the DSS1 interaction activity and no difference between RPA and RPA^{S384A} are observed (Sup. Fig.16F). These data are now included in Supplemental Fig. 16. These data suggest that the observed differences are likely due to the effect of phosphorylation at Ser-384 and not attributable to any defects in DNA binding properties.

In addition, to address the challenge of what phosphorylated RPA at Ser-384 versus a phosphomimetic RPA (Ser-384 to Asp) would look like, we developed a non-canonical amino incorporation strategy to generate single-site phospho-Ser incorporated RPA at position 384 (Figure below). The recombinantly produced wild-type and RPA^{pSer384} proteins are shown below (panel A). Presence of phosphorylated protein is confirmed through western blotting using the pSer384-RPA70 antibody (panel B). This particular data again highlights the excellent specificity of the pSer-384 antibody used in this study. Similar to the phosphomimetic (RPA^{S384D}), RPA^{pSer384} shows higher intrinsic Trp fluorescence (panel C). RPA^{pSer384} binds ssDNA similar to wild-type and phosphomimetic RPA. RPA^{pSer384} shows reduced binding to DSS1 similar to the phosphomimetic (panel E). Please note that the BLI experiment shown here is done in the absence of ssDNA. These experiments are quite exciting and add confidence to our use of the phosphomimetic in this study to represent the phosphorylated version of RPA at position 384.

Figure. Characterization of pSer-384 RPA. **A)** Coomassie image of RPA and site-specific phospho-Serine-394 incorporated proteins. **B)** Western blotting confirms incorporation of pSer and highlights the excellent specificity of the antibody. **C)** Intrinsic tryptophan fluorescence scans of the proteins show configurational changes for RPA^{pSer384}, similar to RPA-S384D. **D)** ssDNA binding activity measured by following Trp. quenching shows similar binding activity for all three proteins. **E)** BLI experiments were performed by tethering GST-DSS1 to streptavidin optical probes and binding to RPA was captured. Both RPA^{pSer384} and RPA^{S384D} show reduced binding to DSS1, and these experiments were performed in the absence of ssDNA. These experiments show that phospho-Serine carrying RPA

and the S384D phosphomimetic RPA have similar biochemical properties. Please note that the small asymmetric migratory shifts in Panels A and B are due to the placement of the 6x-His tag on RPA70 for the wildtype protein and on RPA32 for pSer384-RPA. Both are positioned on the C-terminal end and have no effect on the various activities of RPA.

These data are not included in the manuscript as a detailed description of the methodology and the necessary controls will be described in a separate manuscript. This study is important for the RPA field, and we will detail the synthesis, cell lines, and the characterization as an independent study. The pSer-RPA was generated using methods and a cell line described recently by the Mehl laboratory (DOI: [10.21769/BioProtoc.4541](https://doi.org/10.21769/BioProtoc.4541))

Additional minor points

1. RPA-ssDNA complexes triggers the ATM/ATR cellular DNA damage checkpoint response  this is too much simplified, ATM will stop being activated once DNA breaks get resected.

We have now re-phrased the sentence as follows: "RPA-ssDNA complexes are important for the activation of ATR signaling response and for a mode of double strand break repair triggered by ATM."

2. page 6: "Aurora B kinase only phosphorylates"  "primarily phosphorylates."

Changed as suggested.

3. Figure 2: please include a phosphatase control when introducing the phospho-specific antibody.

We have included the following control experiments to assess the specificity of phospho-specific antibody: 1) RPA was phosphorylated using *in vitro* kinase assay and phosphorylation was assessed using the phospho-specific antibody. Figure S1c shows that the S384 phospho-specific antibody only recognizes the phosphorylated form of RPA and shows no cross-reactivity with total RPA or with the phosphodead mutant. 2) We also incubated the cell lysate with phosphatase, and we see a marked decrease in phosphorylation indicating that the antibody is specific (Figure S1d). It has to be noted that the cell lysis buffer is not as optimal for robust phosphatase activity. 3) The antibody was also found to recognize only the S384 phosphorylated form of RPA that was generated using noncanonical amino acid approach (Figure B shown above) and 4) S384A knock-in mutants show loss of phospho signal.

4. The sentence at the top of page 9 is confusing, first part compares mutant to wild type, and the second part wild type to mutant. Please unify.

Corrected as suggested.

Reviewer #2

In this study, the authors identified and validated a Serine residue (384) on RPA70 that is phosphorylated by Aurora B kinase during mitosis. They conducted a series of complementary biochemical and cellular experiments to show that phosphorylation at this residue promotes cell viability and recovery from replication stress, higher density RPA filaments and conformational changes that impair binding to DSS1. The authors include a model that suggests phosphorylation of RPA70 likely suppresses homologous recombination during mitosis. The manuscript is very well written, and the study includes critical controls which increase the rigor. Given the multiple roles of RPA in DNA replication and repair, the identification of a novel regulatory site on RPA70 during mitosis should be of broad interest. A few comments that require clarification are below.

1. I commend the authors for including the critical control of homozygous knock-in Ser-384-Ala to validate the specificity of their custom-made pS384-RPA70. However, validation of the Ser-384-Ala genotype should also be included. Also, is the 2nd clone (supplementary Fig 5) also a homozygous knock in? Looks like there is a band with the pS384-RPA708 antibody for this clone (Supplementary Figure 5).

The genotype of CRISPR clones-1 and 2 were validated using next-gen sequencing as indicated in the methods. As per reviewer's suggestion, we have included the NGS raw data and alignments in the source data file.

Yes, clone-2 is also a homozygous knock in. For clone 2, we had used a new batch of antibody that showed strong binding and the phospho-signal was saturated within few seconds for the WT lysates. However, this also enhanced the non-specific background noise in the mutant. We have since optimized the antibody dilution to avoid non-specific binding and have updated pS384-RPA70 blot for clone-2 (Figure S5a). We further validated antibody specificity through the following approaches: 1) RPA was phosphorylated using *in vitro* kinase assay and phosphorylation was assessed using the phospho-specific antibody. Figure S1c shows that the S384 phospho-specific antibody only recognizes the phosphorylated form of RPA and shows no cross-reactivity with total RPA or with the phosphodead mutant. 2) We also incubated the cell lysate with a phosphatase as suggested by reviewer-1 and see a marked decrease in phosphorylation indicating that the antibody is specific (Figure S1d). It has to be noted that the cell lysis buffer is not as optimal for robust phosphatase activity. 3) The antibody was also found to recognize only the S384 phosphorylated form of RPA that was generated using a noncanonical amino acid approach (Figure B shown above).

2. Page 8, paragraph 1. In the last sentence, it is not clear what the authors mean by “genomic stress”. It is possible that Ser-384 is induced by other DNA damaging agents, rather than just a reagent that induces replication stress?

We have revised this sentence to indicate the specific stress that was assayed, and it now states, “replication stress” instead of “genomic stress.”

3. If RPA is more efficient at facilitated exchange (FE) than RPA-S384D, wouldn't that make the unphosphorylated form easier to remodel? I assume this depends on how FE is defined. The last sentence of paragraph 1, page 11 states phosphorylation enables easier remodeling. Please clarify.

Yes, this is correct. The RPA^{S384D} would be easier to remove from the ssDNA and this finding agrees with the 5-fold ssDNA binding affinity for the phosphomimetic. However, we do not yet clearly understand how this would translate towards specific functions in mitosis.

4. Supplementary Fig 6. Quantification of the p53 and gammaH2AX levels in the WT and Ser-384-Ala cells would strengthen the results and rigor. The higher basal level of p53 expression in the mutant is very interesting.

We have quantitated basal p53 and gamma H2AX levels normalized to the loading control and have added the plots in Figure S6e and S6f and have updated the results.

5. Fig. 6, did the authors observe an increase in micronuclei in the Ser-384-Ala mutants? This would be expected given the increase in anaphase bridges, which could lead to breakage.

Yes, we did see an increase in micronuclei in cells undergoing cytokinesis attributed to the lagging chromosomes. But we did not observe a significant fraction of micronuclei in the asynchronous population; instead, we saw an increase in the percentage of dead or dying cells suggesting that the cells experiencing segregation defects are likely dying in G1.

6. Background information in the Discussion could be reduced. However, this section should also include a discussion of why the authors think the Ser-384-Ala cells show more apoptosis, higher basal levels of p53, and greater sensitivity to replication stress than wild type cells.

We have trimmed several sections of the discussion and have added a new section in paragraph 2 to discuss the effects of loss of phosphorylation.

7. Minor comment: page 18, “tailor's” is misspelled.

Corrected.

Reviewer #3

An Aurora B-RPA signaling axis secures chromosome segregation fidelity. In this manuscript, the authors report the regulation of RPA by the mitotic kinase AURKB. The authors show that RPA70 can be phosphorylated *in vitro* by AURKB and only at S384 which is located in a consensus sequence of AURKB. S384-RPA70 was also found to be phosphorylated *in vivo* in several phosphoproteomic studies. *In vivo*, phosphorylation of S384-RPA70 occurs only in mitosis and follows phosphorylation of S10-H3. In the manuscript all data were obtained by analyzing *in vivo* and *in vitro*, the behavior of the phosphomimetic mutant S384D or the mutant S384A that cannot be phosphorylated. Ser-384 is located on the DNA binding domain of the protein but the S383D mutant of RPA70 shows the same ssDNA binding properties of RPA as the wild type protein, only the dynamics of the binding seems to be different. However, the S383D mutant induces a higher density of RPA molecules bound to ssDNA. The S384A mutant affects cell viability with hypersensitivity to DNA damage. The S384D mutant inhibits homologous recombination by preventing recruitment of DSS1-BRCA2.

We thank the reviewer for their thorough analysis and the suggestions/questions. Our corrections and explanations are noted below:

I have two major comments:

1. The involvement of AURKB in S384 phosphorylation *in vivo* was evaluated here using the anti-AURKB inhibitor AZD1152. The use of inhibitors without rescue is always questionable, and the entire manuscript is based on this result. A cleaner approach is needed. A depletion of AURKB by siRNA and a rescue with an AUKB resistant to siRNA is currently used in the field (or the use of an inhibitor and a rescue with a kinase resistant to the inhibitor). This would unambiguously prove the involvement of AURKB in S384-RPA70 phosphorylation *in vivo* and will strengthen the data. Only when this is demonstrated can the authors say or conclude that phosphorylation of RPA70 by AURKB induces... This is mandatory.

We now show that the knockdown of Aurora B using shRNA results in a corresponding decrease in RPA70-S384 phosphorylation without affecting total RPA70 levels. (Figure S2e) We also show that re-expressing Aurora B such that it is not recognized by shRNA, rescues S384-RPA70 phosphorylation (Figure S2f). These results combined with AurkB inhibitor and *in vitro* kinase assay, and MS analysis confirm our findings that Aurora B phosphorylates RPA70 at S384.

Indeed, many sentences (or titles), such as the following, border on over-interpretation

Few examples: (there are many)

As suggested, we have revised many of the figure titles and text for *in vitro* data and have removed Aurora B and instead indicated just the phosphorylation status or sites involved.

Page 6 title

"RPA phosphorylation by Aurora B is specific to mitosis." should be "Phosphorylation of RPA70 at S384 is specific to mitosis."

- Corrected

Page 10

"Thus, we propose that Aurora B phosphorylation of RPA is changing the arrangement of DBD-B and this configurational change promotes higher density of RPA molecules bound on ssDNA." should be: "Thus, we propose that phosphorylation of RPA70 at S384 is changing the arrangement of DBD-B and this configurational change promotes higher density of RPA molecules bound on ssDNA".

- Corrected

Page 10 title

"Aurora B phosphorylation promotes formation of higher density RPA binding to ssDNA". Should be "Phosphorylation of RPA70 at S384 promotes formation of higher density RPA binding to ssDNA."

- Corrected

2. Mitotic phosphorylation of RPA70 is critical for chromosome segregation.

“Intriguingly, when we probed for Ser-10 phosphorylation of Histone H3 (the mitotic marker), we observed a 50% decrease in Ser-10 phosphorylation in the phospho- dead mutant (Figures 6a & b) “.

This is indeed intriguing because 100% Ser10-H3 phosphorylation marks mitosis, a decrease to 50% is a surprise since it is induced only by the presence of S384A-RPA70 or the absence of phosphorytable S384A-RPA70. How can this affect phosphorylation of Ser10-H3? Does it affect AURKB activity? localization? ... Because the phenotype observed is reminiscent to an AURKB defect. For instance, what about the following hypothesis: Would it be possible that Ser-384 must be phosphorylated in mitosis to avoid DDR? If Ser-384 is not phosphorylated, DDR is activated and induces an inhibition of AURKB, then a decrease in Ser10-H3 phosphorylation and classic mitotic defects due to a lack of AURKB activity.

The reviewer has highlighted a potential mechanism of Aurora B regulation that we were also considering since Histone H3 is a substrate of Aurora B. Our results show that indeed, the kinase activity of Aurora B is also down regulated in the S384A mutant cells. We see reduced autophosphorylation at the T232 site of Aurora B indicating that the Aurora B activity is reduced by 40% (Figure 6g, h and i). In addition, phosphorylation of Histone H3, a substrate of Aurora B is also reduced suggesting that the Aurora B activity is reduced in the RPA-S384A mutant cells (Figure 6a and b). This indicates a feedback mechanism wherein Aurora B phosphorylates RPA and RPA in turn enhances Aurora B activity.

The rationale that the effect of RPA phosphorylation on Aurora B activity and segregation is direct is several fold: RPA localizes to the centromere in mitosis especially to the ssDNA observed at R loops (Kabeche, et. al., *Science*, 2018). This study by Lee Zou’s group showed that the R loop-ATR-Chk1 axis maintains Aurora B activity. ATR inhibition reduces Aurora B activity without altering Aurora B centromeric localization. More importantly the ATR-Chk1-Aurora B signaling axis is observed in the absence of DNA damage. ATR activation at R loops is triggered by RPA recruitment to ssDNA in R loops. While Zou’s study places RPA upstream of Aurora B (independent of DDR), our study now shows that this is in essence a feedback mechanism wherein Aurora B-mediated phosphorylation of RPA70 at S384 in turn is needed to maintain Aurora B activity (Figure 6i and Figure 8-models). This suggests a direct effect of RPA on chromosome segregation through regulation of Aurora B.

This feedback effect of RPA phosphorylation on Aurora B activity could be mediated through the ATR-Chk1 axis at centromeres, or it could be through direct regulation of Aurora kinase activity by RPA similar to the direct effect of RPA-ssDNA in stimulating the activity of other kinases such as ATR. We are currently working on uncovering the precise mechanism of how RPA affects Aurora B activity and hope to address it in the near future.

As for involvement of DDR, the DDR observed in the RPA-S384A mutant cells in the unperturbed state is upregulation of p53. Previous studies have shown that there is an Aurora-B and p53 feedback mechanism (Teng C-L, et. al, *Cell Cycle*, 2012). Increased p53 levels leads to activation of FBW7, a ubiquitin ligase that in turn degrades Aurora B. However, we do not observe any changes in total Aurora B levels (Figure 6g) indicating that the basal upregulation of p53 is not sufficient to disrupt Aurora B activity in the mutant cells. This indicates that Aurora B activity is not regulated by p53-DDR in mutant cells.

Based on the new results, we have revised our discussion extensively (first and second paragraph) to address the mechanisms involved. We hope that our future studies will systematically tease out each of the mitotic roles of RPA and the mechanisms involved.

“Thus, these chromosomal defects clearly indicate that Ser-384 phosphorylation of RPA70 is important for segregation of chromosomes in mitosis”.

Well, the authors should investigate/provide a mechanism to explain this because it might be a side effect. I mean Ser-384 phosphorylation of RPA70 might indirectly affect segregation (see above).

Please see our detailed response above.

Reviewer #4

Major concerns:

In the main text (p. 15 onwards) the authors describe that configurational changes bring about uptake or loss of deuterium. This is a misunderstanding of the HDX-MS experiment performed here. Loss should not occur during HDX in relation to the conformational dynamics of your protein system (as you are in high D₂O concentration buffer - >90% D₂O). What is presented here is a differential HDX experiment, where you have the HDX of one state minus the HDX of another state, where the conditions are identical except for the difference in state (which can be a mutation, ligand binding, etc). This needs to be described in the correct way in the text, indeed the authors have used the correct phrasing the figure legend of Fig. 7 (more uptake and less uptake in state X (in this case the mutation)).

We thank the reviewer for their careful read of the manuscript. We edited the passages in question and have chosen more precise language. The passage at the bottom of page 15 now reads:

“To better understand how phosphorylation by Aurora B influences the configurational changes within the multi-domain structure of RPA, we performed hydrogen-deuterium exchange mass spectrometry (HDX-MS) analysis of RPA^{WT} and RPA^{S384D} in the absence or presence of ssDNA (Figure 7). Configurational differences cause changes in uptake of deuterium³⁰. The incorporation of deuterium over time is plotted as a comparison between RPA and RPA^{S384D} (Figure 7, and Supplementary Figures 9 -15). The impact of ssDNA binding on uptake on the two protein forms is also shown.”

We have also changed other areas of the manuscript to this precise language (changes are tracked in the revised ms).

There is a lack of required detail in the methods and supporting information for a reader to assess the quality of the HDX-MS experiments and how they can themselves reproduce them. I would recommend that the authors read the ‘White paper’ on reporting HDX-MS experiments, found here:

<https://www.nature.com/articles/s41592-019-0459-y>

There is a requirement by the community that HDX-MS data meet minimum requirements in their reporting – particularly the requirement for the reporting of:

- Dataset Protein state (one column for each condition: that is, apo, ligand-bound, mutant and others)
- HDX reaction details Labeling conditions, for example, percent D₂O, pH(read), temperature and so on
- HDX time course Listing of what time points were analyzed
- HDX controls Description of HDX control samples analyzed

Describe non-deut controls in methods

- Back-exchange (average) for all peptides measured (model system or studied protein) and the interquartile range of these values
- Number of peptides Description of the number of peptides used for which HDX data were obtained
- Sequence coverage Expressed as the percentage of amides covered by the peptides for which HDX data were obtained
- Average peptide length/redundancy Average peptide length and number of readings for any amide (calculated as the total number of peptides for which HDX data were obtained over the total number of amides)
- Replicates (biological or technical) Number and specification of replicate HDX-MS measurements performed for each condition and deuterium incorporation time point
- Repeatability A quantitative measure of the repeatability of deuterium measurement (for example, the average standard deviation from technical replicate measurements of the deuterium content of all peptides from one or more time points for a single condition)
- Significant differences in HDX A value used as a threshold to represent a significant difference in HDX between examined protein states as based on a quantitative measure of repeatability

We thank the author for pointing out this influential paper. We have now included missing parameters in our methods and in the table below as Supplementary Table 1.

We have not included as estimate of back exchange because we are not calculating rates of exchange as part of our HDX-MS analysis. Our experiments are strictly based on a differential analysis between conditions. However, we do include a comparison of our 24-hour time point (considered to be a fully deuterated native condition) and the theoretical exchange. Data quality in HDX is best determined by the coefficient of variation (CV), which is a metric of precision between experimental replicates. This information is in the table below. CV is a more robust metric than standard deviation because it corrects for peptide length. Our average CV is around 0.1 (10%) which indicates a very high degree of accuracy.

HDX data summary	
Protein State	WT, WT+ssDNA, S384D, S384D+ssDNA
HDX reaction details	diluted 1:10 into deuterated reaction buffer (30 mM HEPES, 200 mM KCl, pD 7.8) at 22°C
Time course	0, 0.008, 0.05, 0.5, 3, 30 h
Controls	diluted into a non-deuterated reaction buffer (30 mM HEPES, 200 mM KCl, pD 7.8 at 22°C)
Number of peptides	RPA14= 11 peptides RPA32=18 peptides RPA70=47 peptides
Sequence Coverage	RPA14=68% coverage RPA32=50% coverage RPA70=52% coverage
Average peptide length/ redundancy	RPA14: average peptide length= 10.5, redundancy =0.115 RPA32: average peptide length =9.4, redundancy = 0.06 RPA70: average peptide length = 12.3, redundancy = 0.07
Experimental Replicates	n=3 for each condition and time point
Repeatability	The average standard deviation from experimental replicates of the deuterium content calculated for each time point and condition
Total exchange reported (as percentage)	RPA14= 44 % exchange RPA32= 42 % exchange RPA70 = 48%
Coefficient of variation for 24h time point averaged	RPA14= 0.112 RPA32= 0.122 RPA70= 0.069

Many of these are missing from the manuscript currently.

Another important aspect that needs to be discussed is the %protein that is predicted to be bound to its ligand (in this case DNA) during those HDX experiments. As the amount of complex in your sample greatly effects the extent (and therefore the level of signal-to-noise) and type of HDX observed. Is the 1:1.2 protein:DNA ratio chosen enough to achieve sufficient complex formation during the entire HDX experiment – especially, as this sample is then diluted 1:10 in deuterated buffer (with no ligand present) which would change the equilibrium during the HDX measurements. Consideration of what populations you have in your HDX solution (free protein and protein in complex) can have a drastic influence on the HDX data and, importantly, the conclusions you can make on the data.

We once again thank the reviewer for their careful read of the manuscript. RPA is an exceptional tight binder to ssDNA and binds in a stoichiometric manner (Fig. 3b,c). The binding affinity is well below $<10^{-10}$ M. Thus, equimolar amounts of RPA and ssDNA are sufficient to form stably bound an uniform 1:1 complex. Nevertheless, we used a ratio of 1:1.2 based on our fluorescence anisotropy experiments which show that a 20% excess of DNA is plenty sufficient to bind all RPA. These same experiments show that dilution will not cause significant disassociation during the HDX experimental time scale given the high affinity stoichiometric interactions. Binding of RPA^{WT} and RPA^{S384D} to (dT)₃₅ or (dT)₉₇ ssDNA oligonucleotides were analyzed using size exclusion chromatography in 1:1 ratios and were shown to stay in complex. Moreover, here we show mass photometry analysis of RPA:ssDNA interactions performed with 10 nM (dT)₃₅ and 10 nM RPA^{WT} and a single 1:1 fully bound complex is captured. Thus, complex dissociation under our HDX-MS experimental conditions is not an issue.

Stoichiometric complex formation between human RPA and ssDNA measured using mass photometry. ssDNA (dT)₃₅ (10 nM) and human RPA (10 nM) behave as a single species. When mixed in 1:1 molar ratio (10 nM each), the RPA-ssDNA complex also behaves as a single species. Data again confirms stoichiometric high-affinity interactions between RPA and ssDNA even at very low concentrations.

Another aspect to consider is oligomerisation: You have shown that DNA binding leads to different oligomer formation, with difference observed between WT and phosphomimetic mutant S384D. However, in the main text, you discuss that the S384D would release the RPA's various domains from its configurationally compacted form to promote RPA-protein interactions through these domains. Could you expand the discussion here to consider what effect the oligomerisation observed could have on the HDX-MS data observed. Does it lead to any difference in HDX kinetics – e.g. EX1 or EXX observed in the raw MS spectra, or is all your data EX2 kinetics?

This is a very good question and one that we also asked ourselves given the high binding affinity for some of the domains. Our analysis of the raw data was completed using the industry standard HDExaminer software package detected. It will look for evidence of EX1, EX2, and bimodal exchange. All of the peptides reported in the HDX-MS analysis conformed to EX2 exchange patterns. We also went through the data manually looking for EX1 and bimodal exchange patterns. No convincing data for these patterns was detected. However, key areas of RPA exhibit difference in deuterium uptake based on mutation or DNA binding. These differences in deuterium incorporation complement the orthogonal techniques in this manuscript to support our model of binding and conformational change.

We also do acknowledge that it is hard to determine how multiple RPA are assembling on the DNA when carrying the phosphomimetic substitution. We are reluctant to think of these as cooperative binding of RPA molecules on the ssDNA and structural evidence would be needed to invoke such models. At the moment, we prefer to interpret the binding of RPA^{S384D} as individual entities and occurs because the binding sites on ssDNA have been made available by the alternate configurations of RPA^{S384D}.

Additionally, what is the differential HDX for the WT or S384D vs. WT or S384D + DNA? This would help to untangle whether the HDX effects observed are due to DNA binding or the mutant solely.

In Figure 7, comparisons of WT vs S384D and WT+DNA vs S384D+DNA at 30 seconds are shown. These two comparisons were selected because the goal was to understand the impact of phosphorylation on unbound and bound protein by using a phospho-mimic. In the main paper we state:

“To better understand how phosphorylation by Aurora B influences the configurational changes within the multi-domain structure of RPA, we performed hydrogen-deuterium exchange mass spectrometry (HDX-MS) analysis of RPA^{WT} and RPA^{S384D} in the absence or presence of ssDNA (Figure 7). Configurational changes bring about changes in uptake of deuterium³⁰, and these HDX changes

(incorporation) are plotted as a comparison between RPA and RPA^{S384D} (Figure 7, and Supplementary Figures 9 -15). These changes were further assessed in the absence or presence of ssDNA”.

In your uptake plots for the 13-25 and 166-190 peptides you observe losses of deuterium, either loss of large deuteration in the earliest time points for 13-25 peptide or varied losses and gains of deuterium for 166-190 peptide. This HDX uptake is irregular to observe for meaningful HDX data, which typically increase – or stays the same level – over HDX time. Additionally, the error bars for 13-25 peptide are also very large for what is expected for technical replicates of HDX data. How do you explain these results? They could be due to experimental factors, for instance carry-over and/or back exchange, or possibly due to changes in the equilibrium of oligomerisation or ligand unbinding over time. Addressing the experimental details of complex and oligomer formation predicted discussed above (and reporting the raw MS spectra for this data) will help to understand whether these results are real or artefactual.

Supplemental Figure 9C and 9H.

We thank the reviewer for the attention to detail. The level of deuterium incorporation cannot decrease over time in our experimental set up. We attribute the observed decreases pointed out by the reviewer to be the result of inaccuracy of the data for low intensity peptides. While we performed all the experiments in triplicate, there can be error at early time points and when exchange is low. Rather than hide this data as is common practice, we have chosen to present all our data. While we have reported these peptides, we do not make any conclusions from the lower quality data. All our conclusions are based on peptides for which the data quality is good and significant differences were observed.

REVIEWERS' COMMENTS

Reviewer #1 (Remarks to the Author):

I thank the authors for their efforts on revising the manuscript and the control experiments that further strongly support their model. I am happy to support the manuscript for publication. Phosphorylation of RPA1 is indeed entirely novel and clearly functionally significant.

Maybe one last thing is a sentence I noted in the abstract that may be rephrased: "Disruption of Ser-384 phosphorylation of RPA70 affects Aurora B activity and leads to defects in chromosome segregation". As written, it suggests that RPA affects Aurora B, which then results in phenotype changes. Rather, the sequence is Aurora B  RPA  chromosome segregation defects, as well demonstrated in the manuscript. Therefore, "Disruption of Ser-384 of RPA70 prevents modification by Aurora B, which leads to defects in chromosome segregation" or similar wording may be better.

Reviewer #2 (Remarks to the Author):

The authors have satisfied my prior concerns.

Reviewer #3 (Remarks to the Author):

This manuscript is a revised version of a manuscript that I previously reviewed. It reports the regulation of RPA by the mitotic kinase AURKB. RPA70 is phosphorylated by AURKB on S384. Its phosphorylation has been observed in vivo in several phosphoproteomic studies. In vivo, phosphorylation of S384-RPA70 occurs only in mitosis and follows phosphorylation of S10-H3. AURKB shRNA eliminates phosphorylation of S384 which is recovered after expression of a shRNA-resistant form of AURKB. Ser-384 is located in the DNA binding domain of RPA70, and its phosphorylation affects its binding dynamics. However, the S383D mutant induces a higher density of RPA molecules bound to ssDNA. The S384A mutant affects cell viability with hypersensitivity to DNA damage. The S384D mutant inhibits homologous recombination by preventing recruitment of DSS1-BRCA2.

The authors corrected their manuscript as requested, answered my questions and added the requested experiments.

I have no further comments. The manuscript is easy to read and the results are very solid.

Translated with www.DeepL.com/Translator (free version)

Reviewer #4 (Remarks to the Author):

Roshan et al., has addressed important points on the HDX-MS reporting and interpretation of their results, with additional experimental evidence for RPA-ssDNA complex stability using mass photometry being much appreciated. To increase the transparency and interpretation of their HDX results then they should have separate colouring for 'No difference' in differential HDX versus 'No coverage' achieved on the proteins structure in Figure 7a-b (currently both are represented as white). As 'no difference' is a result of the experiment and can be interpreted, whereas 'no coverage' is a limitation of the experiment and no conclusions should be made on these regions – this is critical as 50-68% protein coverage was obtained. Additionally, comment of the use of HDExaminer for HDX-MS post-analysis needs to be detailed in the Reporting Summary document. Taken together, I recommend that the manuscript be published in Nature Communications after these minor comments have been addressed.

Nature Communications manuscript NCOMMS-22-38687A

Response to reviewers comments

Reviewer #1 (Remarks to the Author):

I thank the authors for their efforts on revising the manuscript and the control experiments that further strongly support their model. I am happy to support the manuscript for publication. Phosphorylation of RPA1 is indeed entirely novel and clearly functionally significant.

- We thank the reviewer for their support and insights.

Maybe one last thing is a sentence I noted in the abstract that may be rephrased: "Disruption of Ser-384 phosphorylation of RPA70 affects Aurora B activity and leads to defects in chromosome segregation". As written, it suggests that RPA affects Aurora B, which then results in phenotype changes. Rather, the sequence is Aurora B  RPA  chromosome segregation defects, as well demonstrated in the manuscript. Therefore, "Disruption of Ser-384 of RPA70 prevents modification by Aurora B, which leads to defects in chromosome segregation" or similar wording may be better.

- edited as suggested

Reviewer #2 (Remarks to the Author):

The authors have satisfied my prior concerns.

- We thank the reviewer for their support and insights.

Reviewer #3 (Remarks to the Author):

This manuscript is a revised version of a manuscript that I previously reviewed. It reports the regulation of RPA by the mitotic kinase AURKB. RPA70 is phosphorylated by AURKB on S384. Its phosphorylation has been observed in vivo in several phosphoproteomic studies. In vivo, phosphorylation of S384-RPA70 occurs only in mitosis and follows phosphorylation of S10-H3. AURKB shRNA eliminates phosphorylation of S384 which is recovered after expression of a shRNA-resistant form of AURKB. Ser-384 is located in the DNA binding domain of RPA70, and its phosphorylation affects its binding dynamics. However, the S383D mutant induces a higher density of RPA molecules bound to ssDNA. The S384A mutant affects cell viability with hypersensitivity to DNA damage. The S384D mutant inhibits homologous recombination by preventing recruitment of DSS1-BRCA2.

The authors corrected their manuscript as requested, answered my questions and added the requested experiments. I have no further comments. The manuscript is easy to read and the results are very solid.

- We thank the reviewer for their support and insights.

Reviewer #4 (Remarks to the Author):

Roshan et al., has addressed important points on the HDX-MS reporting and interpretation of their results, with additional experimental evidence for RPA-ssDNA complex stability using mass photometry being much appreciated. To increase the transparency and interpretation of their HDX results then they should have separate colouring for 'No difference' in differential HDX

versus 'No coverage' achieved on the proteins structure in Figure 7a-b (currently both are represented as white). As 'no difference' is a result of the experiment and can be interpreted, whereas 'no coverage' is a limitation of the experiment and no conclusions should be made on these regions – this is critical as 50-68% protein coverage was obtained.

- We thank the reviewer for their support and insights. This is a very valuable suggestion. We have edited the Figure panels to highlight the 'no coverage' area (using yellow color) from the 'no change' regions. We will use this scheme in all our studies going forward as this will be immediately interpretable to a reader.

Additionally, comment of the use of HDExaminer for HDX-MS post-analysis needs to be detailed in the Reporting Summary document. Taken together, I recommend that the manuscript be published in Nature Communications after these minor comments have been addressed.

- Reporting summary edited with the necessary information as suggested.